# Self-CriTeach: LLM Self-Teaching and Self-Critiquing for Improving Robotic Planning via Automated Domain Generation

Jinbang Huang [1]  Zhiyuan Li [2]  Yuanzhao Hu [3]  Zhanguang Zhang [1]  Mark Coates [4]  Xingyue Quan [1]
Yingxue Zhang [1]

## Abstract

Large Language Models (LLMs) have recently shown strong promise for robotic task planning, particularly through automatic planning domain generation. However, prior approaches largely treat generated planning domains as planning utilities, which are brittle under imperfect logical states and perception noise, overlooking their potential as scalable sources of reasoning supervision and structured reward signals. At the same time, reasoning LLMs depend on chain-of-thought (CoT) supervision that is expensive to collect for robotic tasks, and reinforcement learning (RL) faces challenges on reward engineering. We propose Self-CriTeach, an LLM self-teaching and self-critiquing framework in which an LLM autonomously generates symbolic planning domains that serve a dual role: (i) enabling large-scale generation of robotic planning problem–plan pairs, and (ii) providing structured reward functions. First, the self-written domains enable large-scale generation of symbolic task plans, which are automatically transformed into extended CoT trajectories for supervised fine-tuning. Second, the self-written domains are reused as structured reward functions, providing dense feedback for reinforcement learning without manual reward engineering. This unified training pipeline yields a planning-enhanced LLM with higher planning success rates, stronger cross-task generalization, reduced inference cost, and resistance to imperfect logical states.

[1] Huawei Noah's Ark Lab [2] University of Toronto [3] University of British Columbia [4] McGill University. Correspondence to: Jinbang Huang <jinbang.huang@h-partners.com>, Zhanguang Zhang <zhanguang.zhang@huawei.com>, Yingxue Zhang <yingxue.zhang@huawei.com>.

*Proceedings of the 43rd International Conference on Machine Learning*, Seoul, South Korea. PMLR 306, 2026. Copyright 2026 by the author(s).

## 1. Introduction

Large Language Models (LLMs) have shown strong potential in robotic task planning due to their reasoning capabilities and cross-task generalization (Huang et al., 2022b;a; Wang et al., 2024; Li et al., 2023; Zhao et al., 2023). However, LLM-based planners often suffer from stochastic outputs and error accumulation over long-horizon tasks, leading to failures. To address these issues, prior work has combined LLMs with symbolic search-based algorithms to improve long-horizon planning robustness (Meng et al., 2024; Hu et al., 2023; Liu et al., 2023). More recently, LLMs have been used to automatically infer planning domains (Oswald et al., 2024; Byrnes et al., 2025; Guan et al., 2023; Han et al., 2024a; Huang et al., 2025b). While effective, these approaches primarily treat inferred planning domains as search utilities, overlooking their potential as scalable sources of reasoning supervision and structured feedback for reinforcement learning (RL) (Dalal et al., 2023; Khodeir et al., 2023). As a result, symbolic planning remains largely external to the learned model, resulting in brittleness when faced with imperfect logical states and perceptual noise. This limitation motivates learning-based planners that internalize symbolic planning structure and exhibit resistance to imperfect or noisy logical state representations.

A natural path toward such internalization is suggested by recent advances in reasoning-oriented LLMs, which have advanced substantially through a combination of chain-of-thought (CoT) supervised fine-tuning (SFT) and RL-based post-training (Wei et al., 2022; Cobbe et al., 2021; Zelikman et al., 2022; Schulman et al., 2017; Shao et al., 2024). This SFT + RL paradigm has emerged as a verified and effective pathway for bootstrapping LLM reasoning capability. However, applying this paradigm to robotic planning remains challenging. First, CoT supervision in robotics typically requires large-scale, high-quality, and manually curated reasoning traces, which are costly and difficult to obtain. Second, RL-based improvement is hindered by the lack of structured and scalable reward functions, as robotic tasks involve long-horizon, combinatorial decision-making processes with inherently sparse rewards. (Kulkarni et al., 2016).

Planning domains offer a promising bridge between these challenges. Prior work has shown that symbolic planning can generate scalable robot task plans (Dalal et al., 2023), and transformation between symbols and languages is effective (Pan et al., 2023b; Han et al., 2024b; Tafjord et al., 2021; Wang et al., 2025). In parallel, the structured nature of symbolic planning domains makes them well suited to serve as systematic dense reward signals for improving model performance. Building on these insights, we posit that LLM self-written planning domains provide a unified solution for both supervision and feedback in robotic planning.

We propose SELF-CRITEACH, a self-teaching and self-critiquing framework that reinterprets LLM self-generated planning domains as data sources and training signals rather than mere planning tools. Specifically, symbolic planning domains in Planning Domain Definition Language (PDDL) format, automatically generated by the LLM, fulfill two roles: (1) generate executable task plans that are transformed into context-rich CoT trajectories for supervised fine-tuning; and (2) serve as structured, dense reward functions that enable self-critiquing and reinforcement learning without manual reward engineering. Our contributions include:

**SELF-CRITEACH Framework:** We introduce SELF-CRITEACH, a novel automated framework that treats LLM self-generated PDDL planning domains as reusable knowledge sources, whose compositional structure enables both scalable generation of planning supervision for self-teaching and structured reward signals for self-critiquing via RL.

**Self-teaching via data generation:** SELF-CRITEACH allows the base LLM to produce validated long-horizon planning datasets that extend beyond its intrinsic planning capacity, and to use this data for SFT without human annotation.

**Automatic symbolic–CoT transformation:** We introduce an automatic CoT generation procedure that translates robot symbolic plans and states into a CoT reasoning trace using the base LLM, and empirically demonstrate the effectiveness of the CoT in self-teaching.

**Self-critiquing with planning domains:** The system reuses the self-generated PDDL planning domains as structured reward functions, enabling post RL training without manual reward engineering.

**Empirical gains:** SELF-CRITEACH produces a planning-enhanced LLM that achieves robust planning performance, stronger cross-task generalization, reduced inference token costs, and resistance to imperfect logical estimation.

## 2. Related Work

**Learning to plan** LLMs have emerged as powerful tools for robotic task planning (Huang et al., 2022b;a; Wang et al., 2024; Chen et al., 2024b; Li et al., 2023; Zhao et al., 2023).

Early work treated LLMs as direct planners, but such approaches struggle with long-horizon dependencies and error accumulation (Sermanet et al., 2024; Driess et al., 2023; Chen et al., 2024a; Wang et al., 2024). Subsequent methods use LLMs to guide symbolic search, improving exploration efficiency while preserving planning completeness (Zhao et al., 2023; Yang et al., 2025b; Meng et al., 2024; Hu et al., 2023; Silver et al., 2024), yet they rely on manually engineered planning domains or search structures, limiting scalability. A complementary line of work studies automatic planning domain generation in PDDL (McDermott et al., 1998), where symbolic world models are learned from data or inferred by LLMs through interaction. Existing approaches either refine partial domains (Diehl et al., 2021; Kumar et al., 2023; Silver et al., 2023; Liang et al., 2024; Athalye et al., 2024; Byrnes et al., 2025; Wong et al., 2023; Liu et al., 2024; Zhu et al., 2024; Huang et al., 2025a), construct domains from natural-language descriptions (Guan et al., 2023; Han et al., 2024a; Oswald et al., 2024), or induce domains directly from demonstration trajectories (Huang et al., 2025b). While these results establish LLMs as capable domain generators, prior work largely treats PDDL domains as planning utilities, overlooking their potential as scalable sources of verified reasoning data. Motivated by recent evidence that PDDL can supervise robot motion learning (Dalal et al., 2023; Khodeir et al., 2023), we propose a self-improving framework that leverages LLM-generated planning domains as training data, yielding substantial performance gains.

**Reasoning LLM and Post-training** Early progress in LLM reasoning primarily relied on few-shot prompting, which are proven effective for simple tasks but struggle with complex multi-step reasoning (Brown et al., 2020). Chain-of-thought (CoT) prompting elicits intermediate reasoning steps, yielding substantial performance gains (Wei et al., 2022; Kojima et al., 2022). Moreover, a range of inference-time strategies, such as self-consistency and tree-based search, have further improved reasoning robustness (Wang et al., 2023a; Yao et al., 2023). Beyond prompting, supervised fine-tuning (SFT) has been widely adopted in model training (Ouyang et al., 2022; Cobbe et al., 2021). In particular, correctness-validated and self-refined rationales have led to significant improvements (Zelikman et al., 2022; Yuan et al., 2023; Tong et al., 2024; Lee et al., 2025; Hosseini et al., 2024). More recently, reinforcement learning has emerged as a key component of reasoning-oriented post-training (Schulman et al., 2017; Shao et al., 2024; Achiam et al., 2017), with combined SFT+RL pipelines demonstrating superior performance (Christiano et al., 2017; Ouyang et al., 2022; Rafailov et al., 2023). Symbolic logic has been explored as a means of enhancing the reasoning capacities of LLMs. Prior work has investigated symbol–language transformations to enable logical reasoning in language models (Pan et al., 2023b; Han et al., 2024b; Olausson et al.,

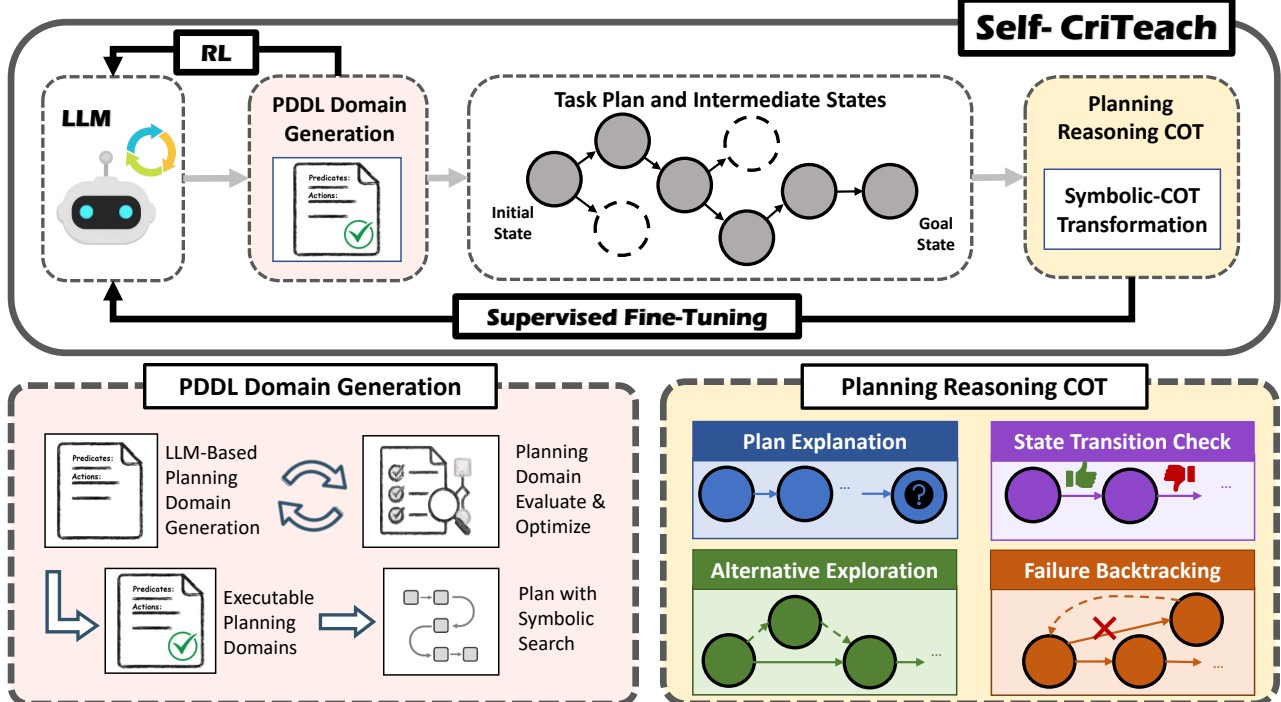

*Figure 1.* Overview of the proposed SELF-CRITEACH framework. The base LLM first generates and iteratively refines PDDL planning domains, which are used to perform symbolic search and produce task plans with intermediate states. These plans are converted into chain-of-thought traces by the same base LLM via plan explanation, state-transition checking, alternative path exploration, and failure backtracking. The resulting CoT data is used for supervised fine-tuning, after which the same self-written planning domains provide structured reward signals for reinforcement learning. Together, supervised and reinforcement learning enable the model to internalize symbolic planning behavior, yielding a reasoning-enhanced LLM with improved generalization and long-horizon planning.

2023; Xu et al., 2024; Pan et al., 2023a; Liu et al., 2023; Tafjord et al., 2021). However, symbolic reasoning traces are often difficult for LLMs to interpret directly (Wang et al., 2025; Feng et al., 2024), particularly in robotics, where physical constraints are expressed in symbolic form. Recent studies further reveal a misalignment between symbolic traces and natural-language reasoning (Stechly et al., 2024), limiting their effectiveness as direct supervision for LLM training. To bridge this gap, we propose an automatic symbolic-to-CoT transformation method based on LLM self-alignment. We empirically demonstrate that this transformation is critical for transferable learning and improves the planning performance of fine-tuned LLMs compared with using raw symbolic planning trajectories.

## 3. Preliminaries

In planning domain definition language (PDDL), a planning domain is defined by $\mathcal{D} = (\mathcal{P}, \mathcal{A})$, where $\mathcal{P}$ is a set of predicates and $\mathcal{A}$ is a set of actions. The object set is defined as $\mathcal{O} = \{o_1, \ldots, o_n\}$. Each **predicate** $p \in \mathcal{P}$ describes object properties or relations characterized by a Boolean classifier $p(o_1, \ldots, o_i) \rightarrow \{0, 1\}$. Instantiating $p$ with objects $o_1, \ldots, o_i \in \mathcal{O}$ yields a ground atom. Let,

$$\mathcal{G} = \{\, p(o_1, \ldots, o_k) \mid p \in \mathcal{P},\ o_i \in \mathcal{O} \,\} \qquad (1)$$

be the set of all possible ground atoms. A symbolic state is a set of true atoms, $\mathcal{X} \subseteq \mathcal{G}$. An **action** $a \in \mathcal{A}$ is defined as $a = \langle \text{PRE},\ \text{EFF}^+,\ \text{EFF}^- \rangle$, where PRE denotes the predicates that must hold for the action to be applicable. The effects consist of add effects $\text{EFF}^+$ and delete effects $\text{EFF}^-$, which specify how the state is updated when the action is executed. Instantiating $a$ with concrete objects $o_1, \ldots, o_j \in \mathcal{O}$ results in a ground action $a(o_1, \ldots, o_j)$. Executing $a$ induces the state transition $\mathcal{X}^t \times a \rightarrow \mathcal{X}^{t+1}$. Thus, a formal definition of a **planning problem** becomes,

$$\mathcal{Q} = \langle \mathcal{O}, \mathcal{D}, \mathcal{X}^{(init)}, \mathcal{X}^{(goal)} \rangle, \mathcal{D} = (\mathcal{P}, \mathcal{A}), \qquad (2)$$

and a solution plan is a sequence of ground actions

$$\tau = \{a^{(0)}, \ldots, a^{(T-1)}\} = \text{PDDLSolver}(\mathcal{Q}), \forall a^{(i)} \in \mathcal{A} \qquad (3)$$

such that $\mathcal{X}^{(init)} \times \tau \rightarrow \mathcal{X}^{(goal)}$.

### 3.1. Automatic LLM Planning Domain Generation

Prior work has demonstrated that a base LLM $\mathcal{M}_0$ is capable of automatically generating planning domains from

unstructured inputs such as natural language descriptions, task specifications, or demonstrations (Han et al., 2024a; Silver et al., 2023; Kumar et al., 2023; Huang et al., 2025b; Oswald et al., 2024; Guan et al., 2023). Formally, we have

$$\hat{\mathcal{D}} \triangleq (\hat{\mathcal{P}}, \hat{\mathcal{A}}), \quad (\hat{\mathcal{P}}, \hat{\mathcal{A}}) = \Psi^{\mathcal{M}_0}(\mathcal{U}), \tag{4}$$

where $\mathcal{U}$ denotes the input source and $\hat{\mathcal{D}}$ is the generated domain with predicates $\hat{\mathcal{P}}$ and actions $\hat{\mathcal{A}}$. $\Psi$ indicates the selected domain generation method. The resulting domain enables planning problems to be solved by symbolic search. Task plans are validated in simulation under physical constraints, assuming a predefined robot skill library.

### 3.2. From PDDL to CoT Generation

Given a generated domain $\hat{\mathcal{D}}$, the symbolic search naturally induces structured reasoning traces. A solution plan $\tau = \{a^{(0)}, \ldots, a^{(T-1)}\}$ encodes a verifiable sequence of state transitions that can be expanded into stepwise natural-language explanations.

**Symbolic trace.** For a planning problem $\mathcal{Q} = \langle \mathcal{O}, \hat{\mathcal{D}}, \mathcal{X}^{(init)}, \mathcal{X}^{(goal)} \rangle$ and its solution $\tau$, each ground action $a^{(t)}$ yields the ordered symbolic state transition trace

$$\mathcal{T}^{sym} = \{(\mathcal{X}^t, a^{(t)}, \mathcal{X}^{t+1})\}_{t=0}^{T-1}. \tag{5}$$

**Natural-language trace.** Each symbolic state transition trace $(\mathcal{X}^t, a^{(t)}, \mathcal{X}^{t+1})$ is then mapped by the base model $\mathcal{M}_0$ into a natural-language explanation:

$$e^{(t)} = f_{\text{NL}}^{\mathcal{M}_0}\left(\mathcal{X}^t, a^{(t)}, \mathcal{X}^{t+1}\right). \tag{6}$$

**CoT trajectory.** Concatenating these explanations yields the full CoT trajectory

$$\text{CoT}_\tau = \{e^{(0)}, e^{(1)}, \ldots, e^{(T-1)}\}, \tag{7}$$

which explicitly aligns symbolic planning semantics with natural-language reasoning.

## 4. Problem Setting

We address the problem of improving a base language model $\mathcal{M}_0$ into a planning-enhanced model $\mathcal{M}_{SCT}$ through the SELF-CRITEACH framework. Given domain inference input $\mathcal{U}$, the base model $\mathcal{M}_0$ induces a symbolic planning domain $\hat{\mathcal{D}}$, on which a symbolic solver generates problem–plan pairs $\langle \mathcal{Q}, \tau \rangle$. Each plan $\tau$ is transformed into CoT explanations $\text{CoT}_\tau$ by $\mathcal{M}_0$. The paired data are then concatenated into full reasoning traces $\zeta_{align} = \langle \mathcal{Q}, \tau, \text{CoT}_\tau \rangle$ which serve as training sources. Aggregated over tasks, the collection $\mathcal{C} = \{\zeta_{align}^i\}_{i=1}^N$ forms a corpus for model fine-tuning. Combining $\mathcal{C}$ for SFT and $\hat{\mathcal{D}}$ as an RL reward signal, the system yields a model $\mathcal{M}_{SCT}$ with improved planning ability, stronger generalization, reduced inference cost, and resistance to imperfect logical states.

## 5. Methodology

As shown in Figure 1, SELF-CRITEACH is a self-teaching and self-critiquing framework that uses a base LLM $\mathcal{M}_0$ to generate and iteratively refine PDDL planning domains, which serve as scalable sources of verified supervision and structured reward signals. This section explains the methodology of SELF-CRITEACH in four stages.

### 5.1. Automatic Planning Domain Generation

The first stage of SELF-CRITEACH focuses on automatically inducing robotic planning domains grounded in physical constraints, building on prior milestone approaches in domain induction. Given a robot task demonstration $\mathcal{U}$, the base LLM $\mathcal{M}_0$ infers predicates and action schema by summarizing simulated robot–object interactions and compiles them into a PDDL planning domain $\mathcal{D} = \langle \hat{\mathcal{P}}, \hat{\mathcal{A}} \rangle$, following the domain generation framework of Huang et al. (2025b). While effective, this framework treats the induced domain as a one-shot output and lacks a feedback mechanism for continual refinement. To address this limitation, we introduce an explicit closed-loop domain correction procedure that incorporates feedback-driven refinement methods from Oswald et al. (2024); Han et al. (2024a). The generated domain is validated on sampled planning problems, and planner failure traces are lifted into structured diagnostic signals for $\mathcal{M}_0$ to perform targeted logical repairs. However, such feedback mechanisms primarily aim to restore feasibility rather than optimize domain compactness. Therefore, we additionally employ a hill-climbing algorithm over the domain structure to prune redundant components while preserving solvability (Silver et al., 2023; Kumar et al., 2023). This combined validation–repair–pruning loop yields compact, executable planning domains suitable for downstream learning. Additional details are provided in Section A.1.

### 5.2. Symbolic-CoT Transformation for Training

The next step transforms symbolic plans into chain-of-thought (CoT) representations by eliciting planner decision reasoning in natural language. This step is essential, as directly training LLMs on raw symbolic structures empirically leads to solution memorization and unstable generalization. Given a planning problem $\mathcal{Q}$ and its solution plan $\tau$, the base model $\mathcal{M}_0$ is prompted with a designed template that converts symbolic state–action transitions into grounded reasoning traces in four aspects. **Plan explanation** prompts the model to explicitly justify why each action is selected in terms of goal progression to expose the intermediate decision structure of symbolic search. **State transition checking** requires the model to verify that constraints, action preconditions, and resulting effects are correctly satisfied, enforcing global plan consistency through step-wise validity checks. **Alternative exploration** asks the

model to enumerate other applicable actions at each state and reason about their potential effects and why they are not prioritized. **Failure backtracking** elicits reasoning over infeasible branches by tracing constraint violations back to earlier decisions. This structured elicitation converts symbolic plans into decision-centric CoT traces, enabling the model to internalize symbolic planning dynamics within its latent reasoning space and improving planning performance through fine-tuning.

To improve robustness and avoid overfitting to a single solution pattern, the planning domain is allowed to generate diverse valid solutions for the same problem and validate CoT correctness via majority voting. This augmentation exposes the model to varied planning strategies, reflecting trade-offs among plan optimality, reasoning depth, and error recovery behaviors. The resulting dataset therefore captures a broad spectrum of structured reasoning patterns, providing rich supervision signals for SFT. Details of the prompt template are provided in Section A.3.

### 5.3. Supervised Fine-Tuning

Each planning tuple comprises a planning problem, its symbolic plan, and the aligned CoT trace $\zeta = \langle \mathcal{Q}, \tau, \mathrm{CoT}_\tau \rangle$, and the full training dataset is $\mathcal{C} = \{\zeta^i\}_{i=1}^N$. During SFT, the model is trained to generate both the action sequence $\tau$ and the explanatory trajectory $\mathrm{CoT}_\tau$ conditioned on the input problem $\mathcal{Q}$. The training objective is the standard autoregressive language modeling loss:

$$\mathcal{L}_{\mathrm{SFT}} = -\sum_{i=1}^{N}\sum_{t=1}^{T_i} \log P\big(y_{i,t} \mid \mathcal{Q}_i; \theta\big), \qquad (8)$$

where $T_i$ is the length of the supervised output sequence for the $i$-th instance, $y_{i,t}$ denotes the $t$-th token of the concatenated plan and reasoning trace $\langle \tau_i, \mathrm{CoT}_{\tau_i} \rangle$, and $\theta$ denotes the model parameters. As shown in prior studies (Ouyang et al., 2022), this process compels the model to produce accurate plans while producing coherent reasoning chains imitating the planning behavior.

### 5.4. Reinforcement Learning

The supervised fine-tuned model $\mathcal{M}_{\mathrm{SFT}}$ is further optimized through reinforcement learning using the self-generated planning domain as a structured reward signal. Unlike sparse success-based rewards, the planning domain provides fine-grained failure feedback, including precondition violations and goal mismatches, enabling step-level plan evaluation instead of binary success/failure signals.

We primarily adopt Constrained Policy Optimization (CPO) (Achiam et al., 2017), which formulates the policy update as a constrained optimization problem:

$$\pi_{k+1} = \arg\max_{\pi} \; \mathbb{E}_{\tau \sim \pi}\left[\sum_{t=0}^{T} R(\mathcal{X}_t, a_t)\right]$$
$$\text{s.t. } \mathbb{E}_{\tau \sim \pi}\left[\sum_{t=0}^{T} C(\mathcal{X}_t, a_t)\right] \leq d, \; D_{\mathrm{KL}}(\pi \,\|\, \pi_k) \leq \delta \qquad (9)$$

where $\pi_k$ denotes the policy at iteration $k$, $d$ is the constraint threshold, $D_{\mathrm{KL}}(\cdot\|\cdot)$ is the KL-divergence between policies, and $\delta > 0$ controls the maximum step size of each policy update. The step-level reward $R(\mathcal{X}_t, a_t)$ quantifies goal predicate satisfaction at state $\mathcal{X}_t$:

$$R(\mathcal{X}_t, a_t) = \frac{|\mathcal{X}^{(goal)} \cap \mathcal{X}_t|}{|\mathcal{X}^{(goal)}|} \qquad (10)$$

where constraint cost $C(\mathcal{X}_t, a_t)$ penalizes actions that violate domain preconditions or produce inconsistent states:

$$C(\mathcal{X}_t, a_t) = \mathbf{1}[\mathrm{prec}(a_t) \not\subseteq \mathcal{X}_t] + \lambda \cdot \mathbf{1}[\neg\,\mathrm{valid}(\mathcal{X}_{t+1})] \quad (11)$$

where $\mathrm{prec}(a_t)$ denotes the preconditions of action $a_t$, $\mathrm{valid}(\cdot)$ checks symbolic consistency with the domain $\mathcal{D}$, and $\lambda$ is a weighting coefficient.

## 6. Experimental Setup

### 6.1. Data and Evaluation Metrics

We evaluate our approach on a variety of planning tasks. During evaluation, the model is prompted with the problem description alone, without additional information, prompting skills, or reasoning traces. We now provide details about the training and testing datasets.

**Training Data:** We have adopted the Blocksworld benchmark, with optimal solution lengths normally distributed between 0 and 20 steps (Valmeekam et al., 2023; Liang et al., 2024). `Blocks World Hard` (BW Hard) increases difficulty by extending the planning horizon, featuring solution lengths up to 60 steps. `Blocks World Align` (BW Align) introduces additional actions and orientation-related requirements, with solutions up to 60 steps. We eventually obtained a training dataset size of 5807 for SFT. Full training details are in Section A.3.

**Testing Data:** In addition to the seen task types used in training, during testing we also include unseen tasks, `Prepare Experiment`, `Reorganize Room`, and `Machine Parts Assembly`. This allows us to evaluate the model's generalization capabilities and its transferability to real-world scenarios. These tasks require similar actions but involve more diverse objects, environments, and goals. The solution lengths are uniformly distributed between 0 and 60 steps. In total, the test set contains over 300 unseen objects and 50 furniture types, forming 1,400

novel test experiments. This design enables a thorough evaluation of the model's ability to generalize. Further details are provided in Section A.2.

**Evaluation Metrics:** We adopt two evaluation metrics: 1) The **planning success rate** measures overall planning performance as the ratio of successfully completed tasks to the total number of tasks, following prior works (Garrett et al., 2020; Silver et al., 2023). 2) The **progress score** quantifies the similarity between the goal state and the resulting state after the first invalid action. This metric is designed to capture partial correctness, particularly in very long-horizon tasks where LLMs rarely achieve full success. The details on metric calculation are provided in Section A.2.

## 6.2. Implementation

We use the search algorithms from the Fast-Forward library (Hoffmann & Nebel, 2011), with a Python interface provided by (Garrett et al., 2020). Each training trace is defined as $\zeta^{\text{align}} = \langle \mathcal{Q}, \tau, \text{CoT}_\tau \rangle$. To prevent overfitting to narrow prompting styles, $\mathcal{Q}$ is dynamically paraphrased by the base model during data generation (Wang et al., 2023b). The RL algorithms studied in this paper include DPO (Rafailov et al., 2023) and CPO (Achiam et al., 2017). Additional details are provided in Section A.3.

## 6.3. Baseline and Ablation

**Baseline Models:** We utilize Qwen3-4B-Instruct-2507 (denoted as Qwen3-4B) (Yang et al., 2025a) as the SELF-CRITEACH base model. The resulting model, *SCT-4B*, is compared against the base model and other state-of-the-art LLMs of comparable scale. These include a larger variant from the Qwen3 family, Qwen3-8B, and other open-source models such as Mistral-24B, Ministral-8B (Liu et al., 2026), Gemma3-12B, Gemma3-4B (Gemma Team, 2025), as well as closed-source models such as GPT-4o (Hurst et al., 2024). Details on baseline model selection and implementation are provided in Section A.2.

**Baseline Approaches:** We additionally compare our method against several reasoning-enhancement techniques for LLM-based robotic planning. These baselines include robotic knowledge distillation (Hinton et al., 2015), instantiated as self-distillation (*Self-Distill*) and teacher–student distillation (*30B-Distill*), where we distill from Qwen3-30B to Qwen3-4B. We further consider prompted chain-of-thought (*Prompt-CoT*) (Wei et al., 2022), which injects explicit reasoning prompts to guide multi-step planning, and *Majority Voting* (Wang et al., 2023a), which aggregates multiple sampled plans and selects the most consistent trajectory. All baseline approaches share the same backbone Qwen3-4B.

**Ablation Studies:** We conduct ablation studies to isolate the contributions of individual components in SELF-CRITEACH. We first assess the role of symbolic-CoT transformation by comparing SCT-4B with $\text{SCT}_{\text{Symbol}}$-4B, which trains directly on symbolic traces without CoT conversion. We further evaluate variants using only supervised fine-tuning ($\text{SCT}_{\text{SFT}}$) or only reinforcement learning. For RL-only variants, we compare our CPO-based training objective ($\text{SCT}_{\text{CPO}}$) with Direct Preference Optimization ($\text{SCT}_{\text{DPO}}$) (Rafailov et al., 2023) and Longest Contiguous Common Subsequence reward ($\text{SCT}_{\text{LCCS}}$) (Huang et al., 2025c), an RL strategy that rewards partial planning progress rather than only complete plan correctness. Comparisons against the full pipeline show that jointly combining symbolic-CoT transformation, SFT, and RL yields the most consistent and robust improvements.

## 7. Results

In this section, we present our experimental results to address the following research questions:

**RQ1.** Can SELF-CRITEACH enable a base LLM to enhance its own planning capabilities? **RQ2.** How does the planning-enhanced model compare with similar-size SOTA models and baseline approaches? **RQ3.** Does the enhanced model demonstrate stronger performance on unseen task types? **RQ4.** Can SELF-CRITEACH advance thinking efficiency?

**RQ1 Effect of Self-CriTeach** The results demonstrate that SCT-4B exhibits substantially stronger planning capability than its base model, particularly in terms of generalization and long-horizon reasoning. As shown in Table 1, SCT-4B achieves consistent improvements in both overall success rate and progress score on unseen tasks, indicating enhanced robustness beyond the training distribution. SCT-4B attains a 20% absolute gain in overall success rate over the base model Qwen3-4B, with an improvement of 21% on `BW Hard` benchmark, highlighting its improved ability to reason over extended planning horizons. Task-specific progress scores are reported in Section A.4.

**RQ2 Comparison to Other Models and Approaches.** We compare SCT-4B against top-performing open-source LLM baselines of similar scale in Table 1. Despite operating at significantly smaller model scales, SCT-4B consistently outperforms all baselines. This performance advantage is especially pronounced on long-horizon tasks such as `BW Hard` and on unseen task distributions. A similar trend is observed when comparing alternative training and inference baselines in Table 2, indicating that SCT-4B's gains arise from stronger planning capability rather than task-specific memorization. While some baselines achieve moderate progress scores, their low success rates reveal difficulties in maintaining global plan consistency. In contrast, SELF-CRITEACH enables SCT-4B to sustain coherent long-horizon planning, resulting in more reliable task completion.

*Table 1.* Planning success rate and progress score across tasks for SCT and SOTA baselines of similar size. The best results are highlighted in bold, second best are underlined. Superscripts show improvement, decline, or no change relative to SCT-4B.

| Model | Seen Tasks Success Rate | | | Unseen Tasks Success Rate | | | Overall | |
|---|---|---|---|---|---|---|---|---|
| | BW Classic | BW Hard | BW Align | Prepare Experiment | Reorganize Room | Machine Parts Assembly | Success Rate | Progress Score |
| SCT-4B (ours) | **0.60** | **0.45** | **0.75** | **0.45** | 0.18 | **0.50** | **0.46** | **0.76** |
| Qwen3-8B | 0.48$^{-0.12}$ | 0.28$^{-0.17}$ | 0.69$^{-0.06}$ | 0.33$^{-0.12}$ | **0.19**$^{+0.01}$ | 0.40$^{-0.1}$ | 0.35$^{-0.11}$ | 0.68$^{-0.08}$ |
| Qwen3-4B | 0.41$^{-0.19}$ | 0.24$^{-0.21}$ | 0.42$^{-0.33}$ | 0.24$^{-0.21}$ | 0.12$^{-0.06}$ | 0.34$^{-0.16}$ | 0.26$^{-0.2}$ | 0.59$^{-0.17}$ |
| Mistral-24B | 0.21$^{-0.39}$ | 0.11$^{-0.34}$ | 0.71$^{-0.04}$ | 0.18$^{-0.27}$ | 0.10$^{-0.08}$ | 0.12$^{-0.38}$ | 0.21$^{-0.25}$ | 0.49$^{-0.27}$ |
| Ministral-8B | 0.03$^{-0.57}$ | 0.02$^{-0.43}$ | 0.05$^{-0.7}$ | 0.01$^{-0.44}$ | 0.02$^{-0.16}$ | 0.02$^{-0.48}$ | 0.02$^{-0.44}$ | 0.14$^{-0.62}$ |
| Gemma3-12B | 0.09$^{-0.51}$ | 0.08$^{-0.37}$ | 0.14$^{-0.61}$ | 0.06$^{-0.39}$ | 0.04$^{-0.14}$ | 0.11$^{-0.39}$ | 0.08$^{-0.38}$ | 0.56$^{-0.2}$ |
| Gemma3-4B | 0.01$^{-0.59}$ | 0.01$^{-0.44}$ | 0.01$^{-0.74}$ | 0.01$^{-0.44}$ | 0.01$^{-0.17}$ | 0.01$^{-0.49}$ | 0.01$^{-0.45}$ | 0.44$^{-0.32}$ |
| GPT-4o | 0.31$^{-0.29}$ | 0.17$^{-0.28}$ | 0.54$^{-0.21}$ | 0.10$^{-0.35}$ | 0.05$^{-0.13}$ | 0.11$^{-0.39}$ | 0.19$^{-0.27}$ | 0.55$^{-0.21}$ |

*Table 2.* Planning success rate and progress score across tasks for SCT and baseline approaches with Qwen3-4B as the backbone. The best results are highlighted in bold; second-best are underlined. Superscripts show improvement, decline, or no change relative to SCT-4B.

| Model | Seen Tasks Success Rate | | | Unseen Tasks Success Rate | | | Overall | |
|---|---|---|---|---|---|---|---|---|
| | BW Classic | BW Hard | BW Align | Prepare Experiment | Reorganize Room | Machine Parts Assembly | Success Rate | Progress Score |
| SCT-4B (ours) | **0.60** | **0.45** | **0.75** | **0.45** | **0.18** | **0.50** | **0.46** | **0.76** |
| 30B-Distill | 0.50$^{-0.1}$ | 0.31$^{-0.14}$ | 0.74$^{-0.01}$ | 0.23$^{-0.22}$ | 0.16$^{-0.02}$ | 0.49$^{-0.01}$ | 0.36$^{-0.1}$ | 0.54$^{-0.22}$ |
| Majority Vote | 0.46$^{-0.14}$ | 0.26$^{-0.19}$ | 0.49$^{-0.26}$ | 0.30$^{-0.15}$ | 0.15$^{-0.03}$ | 0.39$^{-0.11}$ | 0.32$^{-0.14}$ | 0.66$^{-0.1}$ |
| Self-Distill | 0.45$^{-0.15}$ | 0.23$^{-0.22}$ | 0.44$^{-0.31}$ | 0.25$^{-0.2}$ | 0.13$^{-0.05}$ | 0.35$^{-0.15}$ | 0.28$^{-0.18}$ | 0.62$^{-0.14}$ |
| Prompt-CoT | 0.43$^{-0.17}$ | 0.22$^{-0.23}$ | 0.45$^{-0.3}$ | 0.24$^{-0.21}$ | 0.12$^{-0.06}$ | 0.33$^{-0.17}$ | 0.27$^{-0.19}$ | 0.64$^{-0.12}$ |

One clear trend is that more recent LLMs exhibit a substantially improved ability to understand symbolic structures and demonstrate clear advantages in planning tasks. Earlier models show limited understanding of symbolic representations and fail more planning tasks, despite their larger model scales. In contrast, more recent releases, such as Qwen3-4B, achieve markedly better planning performance. This improvement is likely attributable to two factors: the increasing presence of symbolic data in training corpora, and, more importantly, the enhanced reasoning-oriented training of recent models. The first factor is better exposure to symbolic and code-like structures. For example, the Qwen3 series includes a dedicated pretraining stage on mathematics and code data (Yang et al., 2025a), which benefits code-like planning formalisms such as PDDL (Pallagani et al., 2023). This allows the model to better parse formal predicates, action schemas, preconditions, and effects. The second and more critical factor is reasoning-oriented post-training. Qwen3-4B is trained with CoT distilled from stronger reasoning models, which biases the model toward explicit step-wise reasoning rather than direct answer generation. This is particularly important for planning, where success requires maintaining intermediate states, checking whether preconditions hold before each action, applying action effects consistently, and propagating state changes over long transition chains. In other words, planning failures often arise not from misunderstanding a single predicate, but from losing consistency across sequential symbolic

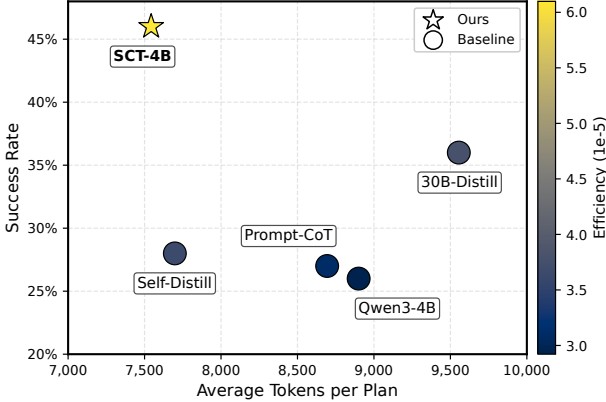

*Figure 2.* Overall success rate versus average per-plan token cost across top-performing baseline approaches.

states. More analysis and observations are presented in the appendix Section A.4.

**RQ3 Generalization** Beyond improvements on seen tasks, SCT-4B demonstrates substantially stronger generalization to unseen tasks, as shown in Table 1. By reusing symbolic representations and replicating internalized planning behaviors across unseen objects, goals, and configurations, SCT-4B achieved transferable planning capabilities that scale robustly across diverse novel scenarios.

**RQ4 Planning Efficiency** To assess planning efficiency, we analyze the ratio between overall success rate and the

*Table 3.* Planning success rate and progress score for SFT-only, RL-only, and symbol-only components compared to base model. The best results are highlighted in bold, second best are underlined. Superscripts show improvement, decline, or no change relative to Qwen3-4B.

| Model | Seen Tasks Success Rate | | | Unseen Tasks Success Rate | | | Overall | |
|---|---|---|---|---|---|---|---|---|
| | BW Classic | BW Hard | BW Align | Prepare Experiment | Reorganize Room | Machine Parts Assembly | Success Rate | Progress Score |
| SCT-4B (ours) | $\mathbf{0.60}^{+0.19}$ | $\mathbf{0.45}^{+0.21}$ | $\underline{0.75}^{+0.33}$ | $\mathbf{0.45}^{+0.21}$ | $\mathbf{0.18}^{+0.06}$ | $\mathbf{0.50}^{+0.16}$ | $\mathbf{0.46}^{+0.2}$ | $\mathbf{0.76}^{+0.17}$ |
| SCT$_{SFT}$-4B | $\underline{0.58}^{+0.17}$ | $\underline{0.41}^{+0.17}$ | $0.67^{+0.25}$ | $\underline{0.42}^{+0.18}$ | $\underline{0.17}^{+0.05}$ | $\underline{0.49}^{+0.15}$ | $\underline{0.43}^{+0.17}$ | $0.67^{+0.08}$ |
| SCT$_{LCCS}$-4B | $0.49^{+0.08}$ | $0.36^{+0.12}$ | $0.51^{+0.09}$ | $0.25^{+0.01}$ | $\underline{0.17}^{+0.05}$ | $0.30^{-0.04}$ | $0.31^{+0.05}$ | $\underline{0.71}^{+0.12}$ |
| SCT$_{CPO}$-4B | $0.52^{+0.11}$ | $0.33^{+0.09}$ | $0.52^{+0.1}$ | $0.29^{+0.05}$ | $\underline{0.17}^{+0.05}$ | $0.35^{+0.01}$ | $0.31^{+0.05}$ | $0.69^{+0.1}$ |
| SCT$_{DPO}$-4B | $0.47^{+0.06}$ | $0.27^{+0.03}$ | $0.49^{+0.07}$ | $0.27^{+0.03}$ | $0.16^{+0.04}$ | $0.36^{+0.02}$ | $0.29^{+0.03}$ | $0.67^{+0.08}$ |
| SCT$_{Symbol}$-4B | $0.54^{+0.13}$ | $0.34^{+0.1}$ | $\mathbf{0.84}^{+0.42}$ | $0.16^{-0.08}$ | $0.14^{+0.02}$ | $\mathbf{0.50}^{+0.16}$ | $0.38^{+0.12}$ | $0.62^{+0.03}$ |
| Qwen3-4B | 0.41 | 0.24 | 0.42 | 0.24 | 0.12 | 0.34 | 0.26 | 0.59 |

*Table 4.* Model planning success rate and progress score before and after SELF-CRITEACH training across different LLM backbones. Superscripts show improvement, decline, or no change relative to the corresponding base model.

| Model | Overall Success Rate | Overall Progress Score |
|---|---|---|
| SCT-Qwen3-8B | $0.49^{+0.14}$ | $0.79^{+0.11}$ |
| Qwen3-8B | 0.35 | 0.68 |
| SCT-Llama-3.1-8B | $0.277^{+0.27}$ | $0.716^{+0.2}$ |
| Llama-3.1-8B | 0.003 | 0.512 |

average token cost per plan. All baseline approaches are included except for majority vote, whose token cost is significantly higher than others. As shown in Figure 2, SCT-4B achieves superior planning performance while maintaining higher efficiency. This improvement stems from the symbolic-CoT transformation, which provides concise supervision by eliminating reasoning steps that do not contribute to planning. Even prompting CoT to follow symbolic state transitions without training already slightly reduces token cost. Notably, SCT-4B outperforms 30B-Distill, indicating that symbolic-CoT supervision offers higher-quality training signals than distilling from large reasoning models. A detailed analysis is provided in Section A.4.

## 8. Ablation Study

In this section, we present ablation studies of SELF-CRITEACH to address the following research questions:

**RQ5.** What are the respective contributions of SFT, RL, and symbolic-CoT in SELF-CRITEACH? **RQ6.** Does SELF-CRITEACH generalize across different LLM backbones?

**RQ5 Roles of Components in SELF-CRITEACH.** As shown in Table 3, using the self-generated planning domain for either SFT or RL alone already significantly improves over the base model. SFT yields substantial gains in success rate across both seen and unseen tasks, indicating that symbolic-CoT supervision provides a strong inductive bias

for structured planning and helps the model internalize complete executable plans. In contrast, RL-based variants show stronger improvements in progress score, suggesting that reward-based optimization is particularly effective at reducing local state-transition violations and improving partial plan generation. This complementary advantage between SFT and RL verifies our design choice of combining both stages in SELF-CRITEACH: SFT improves the likelihood of generating successful complete plans, while RL further reinforces step-wise feasibility and constraint satisfaction.

Among RL variants, CPO consistently outperforms DPO, likely due to CPO's explicit constraint enforcement, which better aligns with the step-wise feasibility checks of symbolic planning. LCCS also outperforms DPO in both success rate and progress score, and matches CPO in overall success rate while achieving a higher progress score. This indicates that optimizing for longer-horizon progress improves partial plan generation, even when full-task success remains challenging. Finally, directly training the base model on symbolic plans without CoT transformation exhibits unstable generalization despite improvements on seen tasks, suggesting a tendency toward solution memorization rather than true internalization of planning structure. This empirically shows that requiring the model to structurally explain the answer before SFT is highly effective for improving generalizability, because this process inherently aligns symbolic representations with the model's reasoning space.

**RQ6 SELF-CRITEACH Backbone Variation.** To verify that our framework is not tied to a specific backbone or model size, we apply SELF-CRITEACH to two additional 8B-scale LLMs: Llama-3.1-8B and Qwen3-8B. As shown in Table 4, both models achieve consistent improvements after SCT training. In particular, Llama-3.1-8B starts with a reasonable progress score but a low success rate, suggesting that it can partially follow planning trajectories but struggles to maintain valid state transitions over multiple steps. After SCT training, this limitation is largely mitigated, leading to a substantial gain in overall success rate. Qwen3-8B also benefits from SCT training, further improving both success

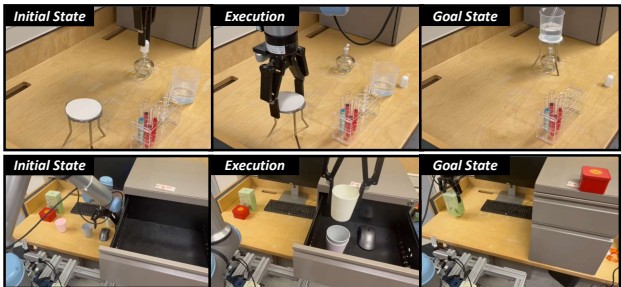

*Figure 3.* Real Robot Planning with SCT-4B: Reorganize Room (Lower, 13 steps) ; Prepare WetLab Experiment (Upper, 8 steps)

*Table 5.* Real-robot task success rates comparison between SCT-4B and PDDL-planner under different perception methods.

| Logical States Estimation | SCT-4B | | PDDL Solver | |
|---|---|---|---|---|
| | **Room** | **Lab** | **Room** | **Lab** |
| Qwen3-VL-4B | 0.70 | 0.60 | 0.40 | 0.20 |
| Rule-based Classifier | 0.80 | 0.90 | 0.70 | 0.70 |

rate and progress score. These results demonstrate that SELF-CRITEACH provides a model-agnostic post-training framework for improving structured planning ability across different LLM backbones.

## 9. Real Robot Experiment

PDDL-based task planners are brittle under incomplete or noisy logical states from imperfect perception. Motivated by this limitation, we conduct two real-robot experiments to evaluate the real-world compatibility of SCT-4B and compare it against a PDDL solver, shown in Figure 3. We deploy SCT-4B on a real UR5e robot using the UR5e control API as low-level skills, and evaluate two tasks, including Reorganize Room (Room) and Prepare WetLab Experiment with water-bath heating (Lab), under two perception pipelines: (1) Low Noise: ground-truth poses with a human engineered rule-based classifier, and (2) High Noise: a VLM, Qwen3-VL-4B (Bai et al., 2025), to directly predict logical states from visual input. Ten trials are performed for each task.

The results in Table 5 demonstrate SCT-4B's improved robustness to imperfect logical states. Both VLMs and rule-based classifiers introduce perception noise: VLMs are affected by partial observability, spatial reasoning errors, and transparent-object detection, while rule-based classifiers can fail in edge cases and nested relations, leading to missing or inconsistent predicates. Under such noisy logical states, PDDL solvers are brittle and prone to failure. Although VLM-based online PDDL state-refinement methods can partially mitigate this brittleness (Liang et al., 2024; Han et al., 2024a), they require additional system-integration effort and remain limited by the VLM's ability to handle transparent

objects or geometrically challenging scenes. In contrast, SCT-4B can reason over and plan with partial symbolic observations despite state-estimation errors. These experiments highlight the stronger deployability of SCT-4B on real robots and its compatibility with practical perception and control pipelines.

## 10. Limitations & Future Work

While SELF-CRITEACH demonstrates strong improvements in planning, several limitations remain. First, it relies on a base LLM with sufficient reasoning and coding ability to induce coherent PDDL domains, as weaker models may fail to produce valid predicates, action schema, or precondition-effect structures. Second, the framework assumes that target tasks can be reasonably abstracted into symbolic planning spaces, which is less suitable for tasks involving continuous constraints, deformable objects, contact-rich manipulation, or fine-grained geometric reasoning. Future work will explore planning-domain generation methods with lower requirements on base model capability. Another important direction is to improve symbolic abstraction for perceptually complex and continuous tasks by learning task-relevant predicates from multi-modal demonstrations. To address the expressivity limitations of symbols, future work can investigate tighter integration with learned low-level controllers, such as VLA models. Finally, we aim to extend SELF-CRITEACH toward adaptive skill discovery and online learning during real-robot execution, allowing robots to identify missing skills, refine action abstractions, and update knowledge from execution failures.

## 11. Conclusion

In this work, we introduce SELF-CRITEACH, a self-teaching and self-critiquing framework that leverages LLM-generated planning domains as scalable sources of verified supervision and structured reward signals. By treating planning domains as data engines, SELF-CRITEACH automatically constructs diverse long-horizon task plans that surpass the base model's intrinsic planning ability without human curation. Through symbolic-to-CoT transformation, the framework bridges formal symbolic structures with natural-language reasoning, enabling models to internalize search behavior and planning skills. RL post-training with the planning domain as rewards further refines planning under structured constraints. Empirically, SELF-CRITEACH yields substantial gains in planning success, generalization, and token efficiency, while maintaining robustness to imperfect logical states during real-robot execution. These results demonstrate its practical applicability to realistic robotic systems and point toward a new paradigm for LLM post-training, contributing to the development of foundation models with stronger long-horizon planning capabilities.

## Impact Statement

This paper presents work whose primary goal is to advance the field of machine learning, particularly in the context of structured planning and reasoning for robotic systems. The techniques developed are intended to improve the robustness, efficiency, and generalization of learning-based planners. There are many potential societal consequences of our work, none of which we feel must be specifically highlighted here. As such, we believe no additional ethical concerns require specific discussion at this time.

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

# A. Appendix

## A.1. LLM-based Domain Generation

**Initial Domain Skeleton Construction**   The first stage of SELF-CRITEACH is the automatic construction of symbolic planning domains. We leverage the generative capacity of the base model $\mathcal{M}_0$ to propose candidate predicates that capture object relations and intrinsic properties, guided by physical simulation. The overall pipeline follows the domain generation framework proposed by Huang et al. (2025b). Input $\mathcal{U}$ is a demonstration trajectory with a short task description. Subsequently, we prompt $\mathcal{M}_0$ with demonstration trajectories collected by the Agilex Pika Data Collection System to invent actions, which are then compiled into executable planning domains. The model outputs a preliminary PDDL domain skeleton $\mathcal{D}_0 = \langle \mathcal{P}, \mathcal{A} \rangle$, where $\mathcal{P}$ denotes the set of predicates and $\mathcal{A}$ the set of actions with preconditions and effects. This skeleton is then tested against a suite of sampled planning problems $\{\mathcal{Q}_i\}$ using the Fast-Forward planner (Hoffmann & Nebel, 2011; Garrett et al., 2020). Successful execution indicates a consistent domain; otherwise, domain errors are detected and used for refinement.

**Feedback–Driven Planning Domain Repair**   When validation fails, SELF-CRITEACH introduces two complementary self-correction mechanisms.

**Feedback prompting.**   Error traces from the planner (e.g.,undefined predicate) are reformulated into feedback prompts. These prompts are re-injected into $\mathcal{M}_0$ to request targeted corrections, following the iterative refinement proposed in (Guan et al., 2023; Oswald et al., 2024).

Formally, given error $e$ produced on problem $\mathcal{Q}_i$, we define a feedback function

$$h(e, \mathcal{Q}_i) \to \text{diagnostic prompt } d,$$

which is appended to the domain-fix query to produce a repaired domain $\mathcal{D}_{t+1}$. This iteration is repeated until a consistent and executable domain $\mathcal{D}'$ is generated. Prompt Template is as following:

---

**Prompt for Domain Error Fixing**

### **Role** ###
You are an expert in AI Planning (PDDL) and robotics task modeling. Your task is to fix mistakes of a PDDL planning domain.

### **PDDL Domain** ###
The current domain is:{Current domain}

### **Problem** ###
The planning problem is: {Planning Problem}

### **Error** ###
An error occurred during solving planning problem, the returned error is:{Error Trace}

---

**Hill-Climbing Search for Domain Redundancy Pruning**   In addition to error repair, generated domains often contain redundant predicates and actions. To address this, we employ a symbolic hill-climbing algorithm (Silver et al., 2023; Kumar et al., 2023; Huang et al., 2025a) that prunes unnecessary components from the domain. This procedure ensures that the final domain $\mathcal{D}^\star$ is both executable and minimal, containing only semantically necessary components.

**Automatic Problem–Plan Pair Generation**   Once a validated domain $\mathcal{D}^\star$ is obtained, we generate a library of problem–plan pairs $(\mathcal{Q}, \tau)$. Each problem is constructed by sampling initial and goal states consistent with $\mathcal{D}^\star$, and solved with the symbolic planner. The resulting pairs are later aligned with chain-of-thought explanations to form the training traces used in SELF-CRITEACH.

## A.2. Evaluation Details

**Evaluation Data Details**    The evaluation dataset consists of seven disjoint task datasets: *stack-200*, *unstack-200*, *reorder-200*, and *align-200* (Blocks World domain), along with *prepare-experiment-200*, *reorganize-room-200*, and *machine-parts-assembly-200*. Each dataset comprises 200 tasks, with solution lengths uniformly sampled across four intervals: 0–10, 10–20, 20–30, and 30+ steps.

Next, we discuss how each testing set is augmented and the detailed difficulty distribution:

**Seen Tasks:**

- *Blocks World-Classic* is a reproduction of the traditional Blocks World benchmark, consisting of 100 problem instances across the stack, unstack, and reorder tasks. The optimal plan lengths approximately follow a normal distribution within 0–20 steps. The details of test problem distribution of Blocks World Classic are shown in Figure 4

- *Blocks World-Hard* extends the benchmark to include more challenging problems with longer optimal plan lengths up to 60 steps. The distribution of problem counts across four difficulty intervals is kept balanced. It contains 200 problem instances for each of the three task types: stack, unstack, and reorder.

- *Blocks World-Align* further extends the benchmark by introducing orientation reasoning. In addition to the standard actions, a rotate action is included, requiring the model to reason about spatial orientations.

**Unseen Tasks**

- *Reorganize Room*: The robot must collect household items, redistribute them to their designated locations, and pack them according to specified requirements.

- *Machine Parts Assembly*: The robot must collect machining parts distributed across the factory and assemble them in the required order.

- *Prepare Experiment*: The robot must retrieve laboratory equipment and set up an experimental platform.

The unseen tasks include a large scale diverse objects (over 300) and furniture(over 50), the details are shown in the following section.

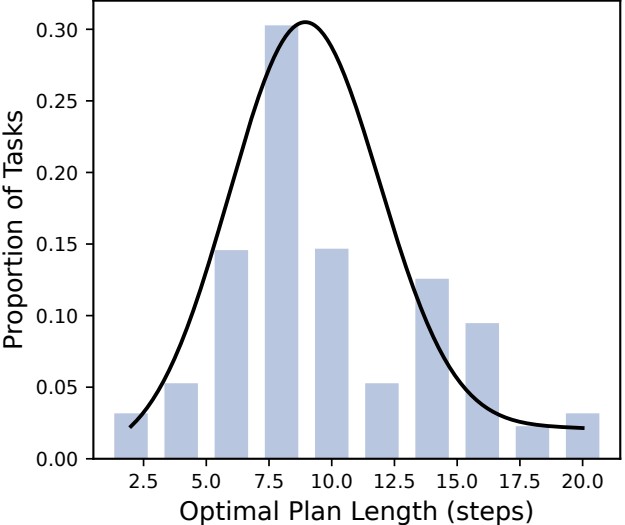

*Figure 4.* Evaluation data distribution for Blocks World Classic

**Unseen Object Types and Furniture Included in Test Set**

### HouseKeeping Objects

shoebox, book, towel, cushion, pillow, blanket, toyblock, jar, canister, bin, basket, tilepack, box, storagebox, detergent, soapbar, tissuebox, magazine, photoalbum, cuttingboard, foodbox, ricebag, flourbag, sugarbag, spicejar, candle, cup, plate, pot, pan, tray, bucket, stepbox, organizer, toybox, craftbox, sewingbox, pillowbox, laundrybox, clothbag, storagecrate, hamper, cushionbox, shelfbox, matpack, drivecase, clipboard, penbox, pencilbox, markerbox, staplebox, tape, tapeholder, calendar, planner, report, documentbox, letterbox, envelopebox, badgebox, tagbox, cardbox, stampbox, inkpad, paperroll, chartbook, whiteboard, pinboard, notepad, scrapbook, catalog, supplybox, lunchbox, laptopbox, headsetcase, monitorbox, keyboardbox, mousebox, cablebox, dockbox, shoes, slippers, sandals, boots, books, magazines, notebooks, comics, albums, photoalbums, towels, napkins, blankets, pillows, cushions, plates, bowls, cups, glasses, mugs, cutlery, forks, spoons, knives, chopsticks, spicejars, condiments, cerealboxes, snackpacks, bottles, jars, cans, storagebins, shoeboxes, laundrybaskets, soapbars, detergents, shampoos, conditioners, lotions, toothbrushes, toothpastes, razors, combs, brushes, hats, scarves, belts, ties, gloves

### Factory Objects

pallet, crate, ingot, brick, block, mold, drum, barrel, tray, spool, battery, foam, plate, rod, beam, sheet, coil, carton, gearbox, motor, casing, bearingbox, brickpack, cablebox, metalbox, plasticbin, boltpail, nutbox, washerbox, pipebundle, timber, lumber, steelbar, rebar, partbox, panel, duct, filterbox, container, powderbag, sack, clampbox, toolkit, spacerblock, fastenerbox, weldrod, fixture, drillbox, pallets, crates, bricks, blocks, beams, pipes, rods, bars, rebars, sheets, panels, plates, coils, rolls, cylinders, drums, barrels, containers, boxes, cartons, bolts, nuts, washers, screws, clamps, wrenches, spanners, drills, toolbits, sockets, filters, gaskets, valves, hoses, cables, chains, belts, wheels, gears, motors, casings, bearings, molds, fixtures, frames, foampads, straps, seals, packaging, labels

### Lab Objects

rack, cylinder, labbox, carton, container, samplebox, tipbox, cryobox, pack, dish, slidebox, capsule, pouch, filterbox, tray, case, testbox, bufferbox, kit, bag, tubecrate, platebox, mediumbottle, sealbag, gelbox, reagentbox, chipbox, cellbox, rackbox, capbox, powderjar, acidbottle, solventcan, stockbottle, samplejar, drybox, packtube, enzymebox, coolerbox, chemcart, bottles, beakers, flasks, cylinders, vials, tubes, testtubes, petri, slides, racks, tipboxes, cryoboxes, samplebags, pipettes, pipettips, dishes, capsules, ampoules, filters, funnels, gloves, masks, goggles, aprons, coats, notebooks, pens, labels, markers, tags, trays, cases, carts, stands, supports, boxes, containers, jars, pouches, packs, media, solutions, buffers, reagents, kits, cells, chips, plates, serums, enzymes

### Housekeeping Furniture

dining table, coffee table, side table, console table, end table, bedside table, kitchen table, foldable table, picnic table, patio table, round table, square table, rectangular table, buffet table, sofa table, low table, tea table, serving table, bench table, counter table, island table, tv stand, hall table, display table, exhibit table, study desk, writing desk, computer desk, standing desk, reception desk, conference table, meeting table, office table, printer stand, workstation, drafting table, blueprint table

### Factory Furniture

workbench, assembly table, packing table, utility table, sorting table, assembly bench, grinding table

### Lab Furniture

lab bench, lab table, specimen table, experiment bench, fume table, inspection table

**Evaluation Implementation Details** We built a unified evaluation pipeline for all experiments. The pipeline loads each evaluation dataset and constructs prompts by combining a system prompt with a task-specific user prompt template. For model inference, a maximum generation length of up to 16,384 tokens is allowed. The tokenizer's built-in chat template is applied to each prompt to ensure consistent formatting. For each model output we extract the final predicted action sequence enclosed in `<FINAL>` tags.

---

**System Prompt for Evaluation**

You are a robot assistant. Your task is to generate a plan given the initial and goal state. A plan is a sequence of actions.

---

**User Prompt for Evaluation**

### **General request**###
Your task is to predict a set of actions that arrive at the goal state starting from the initial state. A state is defined by a set of predicates. Predicates can be static (i.e. describe invariant properties of the environment that do not change over time) or dynamic.

### **Possible Predicates** ### : {Your Predicates}

### **Possible Actions** ###: {Your Actions}

### **Problem to Solve** ###: {Initial State} {Goal State}

### **Output** ###:Always output the final plan inside `<FINAL>` ... `</FINAL>`

---

**Code for Extracting Final Action Sequence**

```python
def extract_answer(output):
    # This pattern ensures no nested <FINAL> inside the capture
    matches = re.findall(r'<FINAL>((?:(?!<FINAL>).)*?)</FINAL>', output, re.DOTALL)
    if matches:
        return re.sub(r'([^,\[\]\s]+)', r'"\1"', matches[-1])  # last match only
    return None
```

---

**Evaluation Metric Details** We evaluated our models in principle on two metrics: **Success Rate** and **Progress Score**. The formal definitions of the metrics follow:

Given a planning problem,

$$\mathcal{Q} = \langle \mathcal{O}, \mathcal{D}, \mathcal{X}^{(\text{init})}, \mathcal{X}^{(\text{goal})} \rangle,$$

and the model's prediction,

$$\tau = \{a^{(0)}, \dots, a^{(T-1)}\},$$

we define,

$$\mathcal{X}_N^{(\text{plan})} = \mathcal{X}^{(\text{init})} \times a^{(0)} \times \cdots \times a^{(N-1)}.$$

Furthermore, we define that for any action $a^{(i)} \notin$ action space of $\mathcal{X}$ (which is an invalid action),

$$\mathcal{X} \times a^{(i)} = \emptyset,$$

and for all $j$,

$$\emptyset \times a^{(j)} = \emptyset.$$

Thus, if the model's predicted plan is valid until $m^{th}$ step, it follows that,

$$m = \min\big(\{\, i \mid \mathcal{X}^{(\text{init})} \times a^{(0)} \times \cdots \times a^{(i)} = \emptyset \,\} \cup \{T\}\big).$$

We define the logical divergence function to describe similarity between 2 states,

$$f_{\text{logical divergence}}(\mathcal{X}^{(i)}, \mathcal{X}^{(j)}) = \frac{|\mathcal{X}^{(i)} \cap \mathcal{X}^{(j)}|}{|\mathcal{X}^{(i)} \cup \mathcal{X}^{(j)}|}.$$

Finally,

$$\textbf{Success Rate}(\mathcal{Q}, \tau) = \mathbf{1}\left[\mathcal{X}_T^{(\text{plan})} \subseteq \mathcal{X}^{(\text{Goal})}\right] \tag{12}$$

$$\textbf{Progress Score}(\mathcal{Q}, \tau) = f_{\text{logical divergence}}(\mathcal{X}_m^{(\text{plan})}, \mathcal{X}^{(\text{goal})}) \tag{13}$$

### A.3. Training Details

**Training Data Details**   The training problems are randomly sampled from the generated PDDL domain, with solution lengths ranging from 0 to 60 steps, resulting in a total of 5,807 examples. Among these, 719 are from `Blocks World Align`, 3,048 from `Blocks World Hard`, and 2,038 from `Blocks World Reorder`. For `Blocks World Hard`, the optimal plan lengths follow a 6:2:1:1 ratio across the intervals 0–10, 10–20, 20–30, and 30+ steps. The symbolic solver is permitted to generate both optimal and suboptimal solutions, allowing the model to learn from shortest-path plans as well as alternative, longer trajectories. During training, an evaluation set is held out, consisting of 80, 340, and 227 examples for the respective task types, corresponding to an evaluation ratio of 0.1.

Each training example, consisting of a problem definition and its corresponding solution (either optimal or suboptimal), is provided to the LLM, which then generates a symbolic-language chain-of-thought alignment. These alignments, together with the problems and solutions, form the ground truth of the training dataset. The system prompt used during training is identical to that used in evaluation. Additionally, we allow dynamic rephrasing of the problem setting in the user prompt during training (relative to evaluation) to help the model maintain focus on the problem context. The details of training problem distribution are shown in Figure 5.

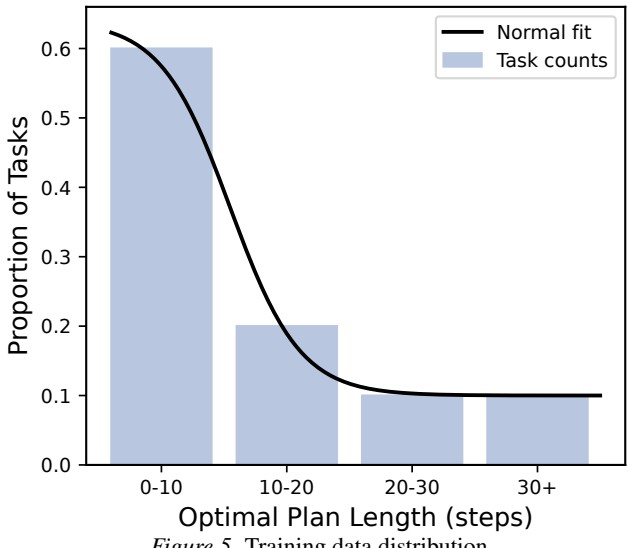

*Figure 5.* Training data distribution

**Training Implementation Details**   The pipeline for generating CoT follows a structure similar to the evaluation pipeline, with minor modifications to the prompts and a maximum generation length of 65,536 tokens. From each model output, we extract the final predicted action sequence enclosed within `<FINAL>` tags. During the supervised fine-tuning (SFT) stage, the base language model is trained to generate valid action sequences conditioned on planning problem descriptions. We train the model for 5 epochs using a learning rate of $1 \times 10^{-5}$ with a per-device batch size of 2, applying gradient accumulation over 2 steps to stabilize optimization under limited batch sizes, and optimize all parameters using Adam. For the reinforcement learning stage, we further optimize the SFT model using Constrained Policy Optimization (CPO) with a learning rate of $1 \times 10^{-6}$, a KL penalty coefficient $\beta = 0.1$, constraint threshold $d = 0.25$, and constraint weight $\lambda = 0.5$. Training is conducted for 3 epochs with an effective batch size of 4 (via gradient accumulation), using the Adam optimizer with gradient clipping at 1.0. During online trajectory sampling, the model generates action sequences with temperature 0.6, top-$p$ 0.95, and a maximum of 16,384 new tokens. Logical feasibility violations where action preconditions are not satisfied by the current state are treated as constraints. Each goal predicate satisfied in the generated state contributes a unit reward, and the total reward is normalized by the number of goal predicates.

---

System Prompt for Generating Symbolic-language Alignment CoT

### Role ###
You are an expert in AI Planning (PDDL) and robotics task modeling. Your task is to generate a detailed chain-of-thought reasoning process for solving the given planning problem.

### Goal ###
You will be provided with:
- The planning domain
- The initial state
- The goal state
- The ground truth task plan

### Task Description ###
Your job is to produce a step-by-step reasoning process that explains:
- Why each action was chosen
- How each action changes the state
- How the evolving state satisfies preconditions and leads toward the goal
- The logical connections between actions, state transitions, and goal achievement
- Explore a few applicable actions at each step other than the provided ground truth
- **After each step, briefly reflect on why alternative actions were not chosen at that point**

### Output ###
You can follow the EXAMPLE reasoning provided, return the full result.

---

User Prompt for Generating Symbolic-language alignment CoT

### Problem Setting: ### {Your Problem Setting}

### Task Description Your task is to explain how to predict a set of actions that arrive at the goal state starting the initial state.

### Task to explain: ###
Initial State: {Your Initial State}
Goal State: {Your Goal State}
Correct plan of actions: {Your Symbolic Plan}

### Solution: ### After your reasoning, put your final explanation in this format:
{<REASON><ANSWER_HERE></REASON> }

---

Code for Extracting CoT Response

```python
def extract_answer(output):
    # This pattern ensures no nested <FINAL> inside the capture
    matches = re.findall(r'<REASON>((?:(?!<REASON>).)*?)</REASON>', output, re.
        DOTALL)
    if matches:
        return matches[-1]  # last match only
    return None
```

## A.4. More Experiment Results and Analysis

Here we provide the detailed success rate, progress score, and average tokens used of each evaluated model under every task type, including some not listed in main body. By analyzing these results, we conclude a few remarks that are not directly related to our approach but meaningful to share.

**Improving Planning Capacities with Recent Models** One clear trend is that more recent LLMs exhibit a substantially improved ability to understand symbolic structures and demonstrate clear advantages in planning tasks. Earlier models, such as Qwen-2.5 and Llama-3, show limited understanding of symbolic representations and fail on most planning tasks, despite their larger model scales. In contrast, more recent releases, including Qwen-3, and Mistral, achieve markedly better planning performance. This improvement is likely attributable to the increasing presence of symbolic data in training corpora, as well as the enhanced reasoning capabilities of newer model architectures.

**Symbolic Planning Structure as CoT**: Symbolic plans provide an effective structural prior for CoT generation. Enforcing symbolic state–transition structure through prompting already reduces token cost and improves planning success, with stronger gains when the model is trained to imitate this structure. SCT-4B achieves both lower token cost and higher success rate. In contrast, CoT distilled from large reasoning models (e.g., 30B-Distill) improves success at the expense of significantly higher token cost. These results show that symbolic-CoT offers a more efficient supervision signal without human curation.

**Complementary Roles of SFT and CPO in Learning Planning Behavior** Integrating SFT with CPO reveals a clear complementary relationship in learning planning behavior. SFT provides a strong structural prior by imitating valid symbolic reasoning patterns, enabling coherent high-level planning, but imitation alone is insufficient to ensure step-wise feasibility over long horizons. Reinforcement learning, particularly via CPO, complements SFT by explicitly enforcing symbolic state-transition constraints, primarily correcting intermediate transition errors. The larger improvement in progress score reflects reduced accumulation of invalid states that often cause late-stage failures. As a result, the combined SFT+CPO training yields plans that are both more successful and more consistently aligned with symbolic legality.

*Table 6.* Planning success rate across tasks for all models. Superscripts show improvement, decline, or no change relative to SCT-4B.

| Model | Seen Tasks Success Rate | | | Unseen Tasks Success Rate | | | Overall |
| | BW Classic | BW Hard | BW Align | Prepare Experiment | Reorganize Room | Machine Parts Assembly | Success Rate |
|---|---|---|---|---|---|---|---|
| SCT-4B (ours) | 0.60 | 0.45 | 0.75 | 0.45 | 0.18 | 0.50 | 0.46 |
| 30B-Distill | $0.50^{-0.1}$ | $0.31^{-0.14}$ | $0.74^{-0.01}$ | $0.23^{-0.22}$ | $0.16^{-0.02}$ | $0.49^{-0.01}$ | $0.36^{-0.1}$ |
| Majority Vote | $0.46^{-0.14}$ | $0.26^{-0.19}$ | $0.49^{-0.26}$ | $0.30^{-0.15}$ | $0.15^{-0.03}$ | $0.39^{-0.11}$ | $0.32^{-0.14}$ |
| Self-Distill | $0.45^{-0.15}$ | $0.23^{-0.22}$ | $0.44^{-0.31}$ | $0.25^{-0.2}$ | $0.13^{-0.05}$ | $0.35^{-0.15}$ | $0.28^{-0.18}$ |
| Prompt-CoT | $0.43^{-0.17}$ | $0.22^{-0.23}$ | $0.45^{-0.3}$ | $0.24^{-0.21}$ | $0.12^{-0.06}$ | $0.33^{-0.17}$ | $0.27^{-0.19}$ |
| $\text{SCT}_{\text{SFT}}$-4B | $0.58^{-0.02}$ | $0.41^{-0.04}$ | $0.67^{-0.08}$ | $0.42^{-0.03}$ | $0.17^{-0.01}$ | $0.49^{-0.01}$ | $0.43^{-0.03}$ |
| $\text{SCT}_{\text{CPO}}$-4B | $0.52^{-0.08}$ | $0.33^{-0.12}$ | $0.52^{-0.23}$ | $0.29^{-0.16}$ | $0.17^{-0.01}$ | $0.35^{-0.15}$ | $0.31^{-0.15}$ |
| $\text{SCT}_{\text{DPO}}$-4B | $0.47^{-0.13}$ | $0.27^{-0.18}$ | $0.49^{-0.26}$ | $0.27^{-0.18}$ | $0.16^{-0.02}$ | $0.36^{-0.14}$ | $0.29^{-0.17}$ |
| $\text{SCT}_{\text{Symbol}}$-4B | $0.54^{-0.06}$ | $0.34^{-0.11}$ | $0.84^{+0.09}$ | $0.16^{-0.29}$ | $0.14^{-0.04}$ | $0.50^{+0.00}$ | $0.38^{-0.08}$ |
| Qwen3-30B | $0.96^{+0.36}$ | $0.70^{+0.25}$ | $0.82^{+0.07}$ | $0.84^{+0.39}$ | $0.47^{+0.29}$ | $0.82^{+0.32}$ | $0.72^{+0.26}$ |
| Qwen3-8B | $0.48^{-0.12}$ | $0.28^{-0.17}$ | $0.69^{-0.06}$ | $0.33^{-0.12}$ | $0.19^{+0.01}$ | $0.40^{-0.1}$ | $0.35^{-0.11}$ |
| Qwen3-4B | $0.41^{-0.19}$ | $0.24^{-0.21}$ | $0.42^{-0.33}$ | $0.24^{-0.21}$ | $0.12^{-0.06}$ | $0.34^{-0.16}$ | $0.26^{-0.2}$ |
| Qwen3-1.7B | $0.07^{-0.53}$ | $0.04^{-0.41}$ | $0.01^{-0.74}$ | $0.00^{-0.45}$ | $0.00^{-0.18}$ | $0.00^{-0.5}$ | $0.02^{-0.44}$ |
| Qwen2.5-7B | $0.02^{-0.58}$ | $0.01^{-0.44}$ | $0.01^{-0.74}$ | $0.00^{-0.45}$ | $0.00^{-0.18}$ | $0.00^{-0.5}$ | $0.00^{-0.46}$ |
| Mistral-24B | $0.21^{-0.39}$ | $0.11^{-0.34}$ | $0.71^{-0.04}$ | $0.18^{-0.27}$ | $0.10^{-0.08}$ | $0.12^{-0.38}$ | $0.21^{-0.25}$ |
| Ministral-8B | $0.03^{-0.57}$ | $0.02^{-0.43}$ | $0.05^{-0.7}$ | $0.01^{-0.44}$ | $0.02^{-0.16}$ | $0.02^{-0.48}$ | $0.02^{-0.44}$ |
| Gemma-3-12b | $0.09^{-0.51}$ | $0.08^{-0.37}$ | $0.14^{-0.61}$ | $0.06^{-0.39}$ | $0.04^{-0.14}$ | $0.11^{-0.39}$ | $0.08^{-0.38}$ |
| Gemma-3-4b | $0.01^{-0.59}$ | $0.01^{-0.44}$ | $0.01^{-0.74}$ | $0.01^{-0.44}$ | $0.01^{-0.17}$ | $0.01^{-0.49}$ | $0.01^{-0.45}$ |
| GPT-4o | $0.31^{-0.29}$ | $0.17^{-0.28}$ | $0.54^{-0.21}$ | $0.10^{-0.35}$ | $0.05^{-0.13}$ | $0.11^{-0.39}$ | $0.19^{-0.27}$ |
| SCT-Llama-8B | $0.43^{-0.17}$ | $0.35^{-0.1}$ | $0.64^{-0.11}$ | $0.03^{-0.42}$ | $0.05^{-0.13}$ | $0.13^{-0.37}$ | $0.28^{-0.18}$ |
| Llama-3.1-8B | $0.01^{-0.59}$ | $0.00^{-0.45}$ | $0.01^{-0.74}$ | $0.00^{-0.45}$ | $0.00^{-0.18}$ | $0.00^{-0.5}$ | $0.00^{-0.46}$ |

*Table 7.* Planning progress score across tasks for all models. Superscripts show improvement, decline, or no change relative to SCT-4B.

| Model | Seen Tasks Progress Score | | | Unseen Tasks Progress Score | | | Overall |
| | BW Classic | BW Hard | BW Align | Prepare Experiment | Reorganize Room | Machine Parts Assembly | Progress Score |
|---|---|---|---|---|---|---|---|
| SCT-4B (ours) | 0.94 | 0.76 | 0.95 | 0.70 | 0.65 | 0.84 | 0.76 |
| 30B-Distill | $0.71^{-0.23}$ | $0.46^{-0.3}$ | $0.92^{-0.03}$ | $0.41^{-0.29}$ | $0.40^{-0.25}$ | $0.68^{-0.16}$ | $0.54^{-0.22}$ |
| Majority Vote | $0.75^{-0.19}$ | $0.59^{-0.17}$ | $0.81^{-0.14}$ | $0.62^{-0.08}$ | $0.55^{-0.1}$ | $0.64^{-0.2}$ | $0.66^{-0.1}$ |
| Self-Distill | $0.71^{-0.23}$ | $0.55^{-0.21}$ | $0.75^{-0.2}$ | $0.57^{-0.13}$ | $0.52^{-0.13}$ | $0.62^{-0.22}$ | $0.62^{-0.14}$ |
| Prompt-CoT | $0.72^{-0.22}$ | $0.57^{-0.19}$ | $0.78^{-0.17}$ | $0.59^{-0.11}$ | $0.54^{-0.11}$ | $0.64^{-0.2}$ | $0.64^{-0.12}$ |
| $SCT_{SFT}$-4B | $0.80^{-0.14}$ | $0.63^{-0.13}$ | $0.91^{-0.04}$ | $0.66^{-0.04}$ | $0.51^{-0.14}$ | $0.74^{-0.1}$ | $0.67^{-0.09}$ |
| $SCT_{CPO}$-4B | $0.85^{-0.09}$ | $0.66^{-0.1}$ | $0.88^{-0.07}$ | $0.61^{-0.09}$ | $0.59^{-0.06}$ | $0.74^{-0.1}$ | $0.69^{-0.07}$ |
| $SCT_{DPO}$-4B | $0.82^{-0.12}$ | $0.63^{-0.13}$ | $0.86^{-0.09}$ | $0.59^{-0.11}$ | $0.58^{-0.07}$ | $0.72^{-0.12}$ | $0.67^{-0.09}$ |
| $SCT_{Symbol}$-4B | $0.79^{-0.15}$ | $0.58^{-0.18}$ | $0.98^{+0.03}$ | $0.44^{-0.26}$ | $0.46^{-0.19}$ | $0.71^{-0.13}$ | $0.62^{-0.14}$ |
| Qwen3-30B | $0.95^{+0.01}$ | $0.72^{-0.04}$ | $0.88^{-0.07}$ | $0.89^{+0.19}$ | $0.56^{-0.09}$ | $0.84^{+0.00}$ | $0.76^{+0.00}$ |
| Qwen3-8B | $0.78^{-0.16}$ | $0.62^{-0.14}$ | $0.93^{-0.02}$ | $0.64^{-0.06}$ | $0.57^{-0.08}$ | $0.72^{-0.12}$ | $0.68^{-0.08}$ |
| Qwen3-4B | $0.73^{-0.21}$ | $0.58^{-0.18}$ | $0.72^{-0.23}$ | $0.52^{-0.18}$ | $0.52^{-0.13}$ | $0.63^{-0.21}$ | $0.59^{-0.17}$ |
| Qwen3-1.7B | $0.58^{-0.36}$ | $0.46^{-0.3}$ | $0.70^{-0.25}$ | $0.37^{-0.33}$ | $0.46^{-0.19}$ | $0.40^{-0.44}$ | $0.47^{-0.29}$ |
| Qwen2.5-7B | $0.51^{-0.43}$ | $0.46^{-0.3}$ | $0.75^{-0.2}$ | $0.43^{-0.27}$ | $0.49^{-0.16}$ | $0.45^{-0.39}$ | $0.50^{-0.26}$ |
| Mistral-24B | $0.51^{-0.43}$ | $0.37^{-0.39}$ | $0.95^{+0.00}$ | $0.35^{-0.35}$ | $0.35^{-0.3}$ | $0.58^{-0.26}$ | $0.49^{-0.27}$ |
| Ministral-8B | $0.22^{-0.72}$ | $0.14^{-0.62}$ | $0.19^{-0.76}$ | $0.07^{-0.63}$ | $0.18^{-0.47}$ | $0.17^{-0.67}$ | $0.14^{-0.62}$ |
| Gemma-3-12b | $0.64^{-0.3}$ | $0.53^{-0.23}$ | $0.83^{-0.12}$ | $0.49^{-0.21}$ | $0.50^{-0.15}$ | $0.53^{-0.31}$ | $0.56^{-0.2}$ |
| Gemma-3-4b | $0.51^{-0.43}$ | $0.43^{-0.33}$ | $0.69^{-0.26}$ | $0.34^{-0.36}$ | $0.41^{-0.24}$ | $0.35^{-0.49}$ | $0.44^{-0.32}$ |
| GPT-4o | $0.68^{-0.26}$ | $0.54^{-0.22}$ | $0.88^{-0.07}$ | $0.41^{-0.29}$ | $0.45^{-0.2}$ | $0.49^{-0.35}$ | $0.55^{-0.21}$ |
| SCT-Llama-8B | $0.84^{-0.1}$ | $0.79^{+0.03}$ | $0.93^{-0.02}$ | $0.46^{-0.24}$ | $0.51^{-0.14}$ | $0.78^{-0.06}$ | $0.72^{-0.04}$ |
| Llama-3.1-8B | $0.54^{-0.4}$ | $0.46^{-0.3}$ | $0.80^{-0.15}$ | $0.45^{-0.25}$ | $0.48^{-0.17}$ | $0.46^{-0.38}$ | $0.51^{-0.25}$ |

*Table 8.* Planning token count across tasks for all models. Superscripts show improvement, decline, or no change relative to SCT-4B.

| Model | Seen Tasks Token Count | | | Unseen Tasks Token Count | | | Overall |
| | BW Classic | BW Hard | BW Align | Prepare Experiment | Reorganize Room | Machine Parts Assembly | Token Count |
|---|---|---|---|---|---|---|---|
| SCT-4B (ours) | 5521 | 8437 | 5298 | 7654 | 8312 | 6341 | 7543 |
| 30B-Distill | $9101^{+3,580}$ | $10785^{+2,348}$ | $5558^{+260}$ | $10624^{+2,970}$ | $10788^{+2,476}$ | $7560^{+1,219}$ | $9555^{+2,012}$ |
| Majority Vote | $21856^{+16,335}$ | $34192^{+25,755}$ | $20834^{+15,536}$ | $31024^{+23,370}$ | $32576^{+24,264}$ | $25712^{+19,371}$ | $29864^{+22,321}$ |
| Self-Distill | $5895^{+374}$ | $8723^{+286}$ | $5412^{+114}$ | $7891^{+237}$ | $8567^{+255}$ | $6589^{+248}$ | $7698^{+155}$ |
| Prompt-CoT | $7234^{+1,713}$ | $9845^{+1,408}$ | $6123^{+825}$ | $9034^{+1,380}$ | $9512^{+1,200}$ | $7856^{+1,515}$ | $8694^{+1,151}$ |
| $SCT_{SFT}$-4B | $5536^{+15}$ | $8448^{+11}$ | $5391^{+93}$ | $7686^{+32}$ | $8348^{+36}$ | $6316^{-25}$ | $7584^{+41}$ |
| $SCT_{CPO}$-4B | $5042^{-479}$ | $6345^{-2,092}$ | $4650^{-648}$ | $5887^{-1,767}$ | $6795^{-1,517}$ | $5610^{-731}$ | $6012^{-1,531}$ |
| $SCT_{DPO}$-4B | $5610^{+89}$ | $8572^{+135}$ | $5413^{+115}$ | $7792^{+138}$ | $8390^{+78}$ | $6203^{-138}$ | $7621^{+78}$ |
| $SCT_{Symbol}$-4B | $7689^{+2,168}$ | $10328^{+1,891}$ | $4483^{-815}$ | $10876^{+3,222}$ | $9908^{+1,596}$ | $7494^{+1,153}$ | $9107^{+1,564}$ |
| Qwen3-30B | $9435^{+3,914}$ | $11418^{+2,981}$ | $9612^{+4,314}$ | $9948^{+2,294}$ | $11923^{+3,611}$ | $10339^{+3,998}$ | $10868^{+3,325}$ |
| Qwen3-8B | $8564^{+3,043}$ | $9368^{+931}$ | $6409^{+1,111}$ | $8413^{+759}$ | $8983^{+671}$ | $8510^{+2,169}$ | $8631^{+1,088}$ |
| Qwen3-4B | $8409^{+2,888}$ | $9963^{+1,526}$ | $6657^{+1,359}$ | $8319^{+665}$ | $9323^{+1,011}$ | $8316^{+1,975}$ | $8929^{+1,386}$ |
| Qwen3-1.7B | $7133^{+1,612}$ | $7440^{-997}$ | $6518^{+1,220}$ | $7268^{-386}$ | $6980^{-1,332}$ | $5917^{-424}$ | $7000^{-543}$ |
| Qwen2.5-7B | $563^{-4,958}$ | $905^{-7,532}$ | $560^{-4,738}$ | $1165^{-6,489}$ | $570^{-7,742}$ | $550^{-5,791}$ | $794^{-6,749}$ |
| Mistral-24B | $894^{-4,627}$ | $1089^{-7,348}$ | $853^{-4,445}$ | $906^{-6,748}$ | $785^{-7,527}$ | $717^{-5,624}$ | $920^{-6,623}$ |
| Ministral-8B | $2871^{-2,650}$ | $3195^{-5,242}$ | $1952^{-3,346}$ | $2707^{-4,947}$ | $1847^{-6,465}$ | $2001^{-4,340}$ | $2547^{-4,996}$ |
| Gemma-3-12b | $582^{-4,939}$ | $704^{-7,733}$ | $563^{-4,735}$ | $732^{-6,922}$ | $550^{-7,762}$ | $617^{-5,724}$ | $653^{-6,890}$ |
| Gemma-3-4b | $765^{-4,756}$ | $926^{-7,511}$ | $772^{-4,526}$ | $953^{-6,701}$ | $772^{-7,540}$ | $909^{-5,432}$ | $883^{-6,660}$ |
| GPT-4o | $2590^{-2,931}$ | $2761^{-5,676}$ | $2790^{-2,508}$ | $2791^{-4,863}$ | $2640^{-5,672}$ | $2796^{-3,545}$ | $2757^{-4,786}$ |
| SCT-Llama-8B | $5912^{+391}$ | $8234^{-203}$ | $5567^{+269}$ | $7823^{+169}$ | $8156^{-156}$ | $6512^{+171}$ | $7534^{-9}$ |
| Llama-3.1-8B | $8234^{+2,713}$ | $9156^{+719}$ | $6387^{+1,089}$ | $8412^{+758}$ | $8876^{+564}$ | $8234^{+1,893}$ | $8523^{+980}$ |

## A.5. Overall Pipeline Pseudocode

---

**Algorithm 1** Full Planning Pipeline

---

1: **Procedure** FULLPLANPIPELINE$(O, M_0, \Psi, U, N)$
2:   $(\hat{P}, \hat{A}) \leftarrow \Psi_{M_0}(U)$                                                {Generate predicates/actions}
3:   $\hat{D} \leftarrow (\hat{P}, \hat{A})$                                                       {Construct domain}
4:   $\mathcal{C} \leftarrow \emptyset$                                                          {Problem–CoT pairs}
5:   **for** $i = 1$ to $N$ **do**
6:     $(X^{init}, X^{goal}) \leftarrow$ RANDOMSAMPLE$(O, \hat{D})$
7:     $Q_i \leftarrow (O, \hat{D}, X^{init}, X^{goal})$
8:     $\tau \leftarrow$ PDDL_SOLVER$(Q_i)$
9:     $\text{CoT}_\tau \leftarrow \{f_{M_0}^{NL}(X^t, a^t, X^{t+1})\}_{t=0}^{T-1}$
10:     $\mathcal{C} \leftarrow \mathcal{C} \cup \{(Q_i, \text{CoT}_\tau)\}$
11:   **end for**
12:   $M_{\text{SFT}} \leftarrow$ SFT$(M_0, \mathcal{C})$                                       {SFT on CoT corpus}
13:   $M_{\text{SCT}} \leftarrow$ RL$(M_{\text{SFT}}, \hat{D})$                          {RL with domain-based reward}
14:   Scores $\leftarrow \emptyset$
15:   **for** $Q_j \in$ TestSet **do**
16:     $\hat{y} \leftarrow M_{\text{SCT}}(Q_j)$
17:     $\hat{\tau} \leftarrow$ EXTRACTPLAN$(\hat{y})$
18:     Scores $\leftarrow$ Scores $\cup \{$PLANVALIDATION$(\hat{\tau}, Q_j)\}$
19:   **end for**
20:   $\text{Score}_{avg} \leftarrow \frac{1}{|\text{Scores}|} \sum \text{Scores}$
21:   **return** $M_{\text{SCT}}, \text{Score}_{avg}$

---

## A.6. Example LLM-generated PDDL Domain

**System Prompt for Generating CoT**

```
(define (domain LLM_generated_domain)
    (:requirements :strips :equality)
    (:predicates
        (obj ?b1)
        (on-table ?b1 ?t1)
        (holding ?b1 ?r1)
        (hand_free ?r1)
        (top ?b2)
        (above ?b1 ?b2)
        (robot ?r1)
        (table ?t1)
        (aligned ?b1 ?b2)
    )

    (:action pick-up
        :parameters (?b1 ?t1 ?r1)
        :precondition (and (obj ?b1) (robot ?r1)(on-table ?b1 ?t1) (top ?b1)
        (hand_free ?r1) (table ?t1))
        :effect (and (not (hand_free ?r1)) (not (on-table ?b1 ?t1)) (holding ?b1
        ?r1))
    )

    (:action stack
        :parameters (?b1 ?b2 ?r1)
        :precondition (and (obj ?b1)(top ?b1) (holding ?b1 ?r1) (robot ?r1)(top
        ?b2) (obj ?b2))
        :effect (and (above ?b1 ?b2) (hand_free ?r1) (not (top ?b2)) (not
        (holding ?b1 ?r1)))
    )

    (:action unstack
        :parameters (?b1 ?b2 ?r1)
        :precondition (and (obj ?b1) (robot ?r1)(top ?b1) (above ?b1 ?b2)
        (hand_free ?r1)(obj ?b2))
        :effect (and (not (hand_free ?r1)) (not (above ?b1 ?b2)) (top ?b2)
        (holding ?b1 ?r1))
    )

    (:action put-down
        :parameters (?b1 ?t1 ?r1)
        :precondition (and (obj ?b1)(holding ?b1 ?r1) (top ?b1) (robot ?r1)
        (table ?t1))
        :effect (and (hand_free ?r1) (not (holding ?b1 ?r1)) (on-table ?b1 ?t1))
    )

    (:action rotate
        :parameters (?b1 ?b2 ?t1 ?r1)
        :precondition (and (table ?t1) (robot ?r1) (obj ?b2) (obj ?b1) (on-table
        ?b2 ?t1) (holding ?b1 ?r1))
        :effect (and (aligned ?b1 ?b2))
    )
)
```

## A.7. Real Robot Experiment Implementation Details

The real robot experiment follows a three-stage pipeline: perception, planning, and execution. First, we extract the initial logical state of the scene using either a VLM or a rule-based classifier. Next, SCT-4B generates a subtask plan. Finally,

the plan is dispatched to the UR5e controller for execution. Throughout the experiment, goal states, object positions, and subtask execution routines are predefined and treated as ground truth.

**Initial State Perception: VLM**    We use Qwen3-4B-VL-Instruct(Bai et al., 2025) as the visual perception module. The model receives a single RGB image of the scene together with a predicate vocabulary and directly outputs a set of PDDL predicates representing the initial logical state. The prompt supplies the image and the list of admissible predicates; the model returns the subset that holds in the depicted scene. An example prompt is shown below.

---

**VLM Perception Prompt**

```
You are given views of a table-top scene with cups.
Describe the state of every object you see within the PDDL domain. This will be
used as the initial state of a pddl problem.

## PDDL Domain ##
A state is defined by a set of predicates.
Possible Predicates in Domain: on-table, holding, hand_free, top, above, beside,
nothing_beside
Possible objects: robot, table, drawer, and some objects
Possible Actions in Domain:
- [pick-up, b, t, r]: take b from table t; requires [top, b], [on-table, b, t],
[hand_free, r]
- [put-down, b, t, r]: place b on table t; requires [holding, b, r]
- [unstack, b1, b2, r]: remove b1 from b2; requires [top, b1], [above, b1, b2],
[hand_free, r]
- [stack, b1, b2, r]: place b1 on b2; requires [holding, b1, r], [top, b2]
- [align, b1, b2, t, r]: place beside on table t; requires [holding, b1, r],
[on-table, b2, t], [nothing_beside, b2]

Initially, the drawer is open, and robot is hand_free.

The goal state of the problem is: [hand_free, robot], [above, mouse, drawer],
[top, mouse], [above, pink_cup, drawer], [above, grey_cup, pink_cup], [above,
white_cup, grey_cup], [top, white_cup], [above, green_box, drawer], [top,
green_box], [above, red_box, drawer], [top, red_box]

You need to provide the initial state based on the image of this problem.
return you final answers in <FINAL>your final predicates</FINAL>
```

---

**Initial State Perception: Rule-based classifier**    As an upper-bound perception baseline we use ground-truth 6-DoF object poses obtained from the robot workspace. Geometric rules map spatial relationships to PDDL predicates: for example, `on-table(obj, tbl)` is asserted when the object's $z$-coordinate is within a threshold of the table surface, and `holding(obj, robot)` is asserted when the gripper is closed around the object.

**Subtask-plan Generation**    The extracted initial state is passed to SCT-4B together with a predefined goal state, using the same evaluation prompt format described in Section A.2. The model generates a chain-of-thought reasoning trace followed by a final action sequence enclosed in `<FINAL>` tags.

**Low-level Execution**    Each PDDL action in the generated plan is mapped to a predefined pick-and-place skill with known object positions. The UR5e arm is controlled via the RTDE library, which sends target joint configurations and Cartesian waypoints to the robot controller.

### A.8. VLA Implementation Details

To further investigate the application of high-level planners in robotic tasks, we integrate our SCT-4B model with the $\pi_0$ model (Black et al., 2024), which serves as the low-level executor, to control a UR5e robot in a set of table-organization tasks within our real-robot experiment.

The SCT-4B model produces high-level commands expressed in PDDL, which are passed to the $\pi_0$ model. The $\pi_0$ model, fine-tuned on 200 real-world pick-and-place trajectories using PDDL-based prompts, translates these plans into low-level spatial delta poses for the UR5e. Execution is then handled by the UR5e's built-in controllers.

The overall task involves picking up and placing objects in different locations to place everything in place and reach a required table-top configuration. This VLA experiment serves as a proof of concept, demonstrating that our high-level planners can be seamlessly connected to a lower-level VLA model to carry out real-world tasks.

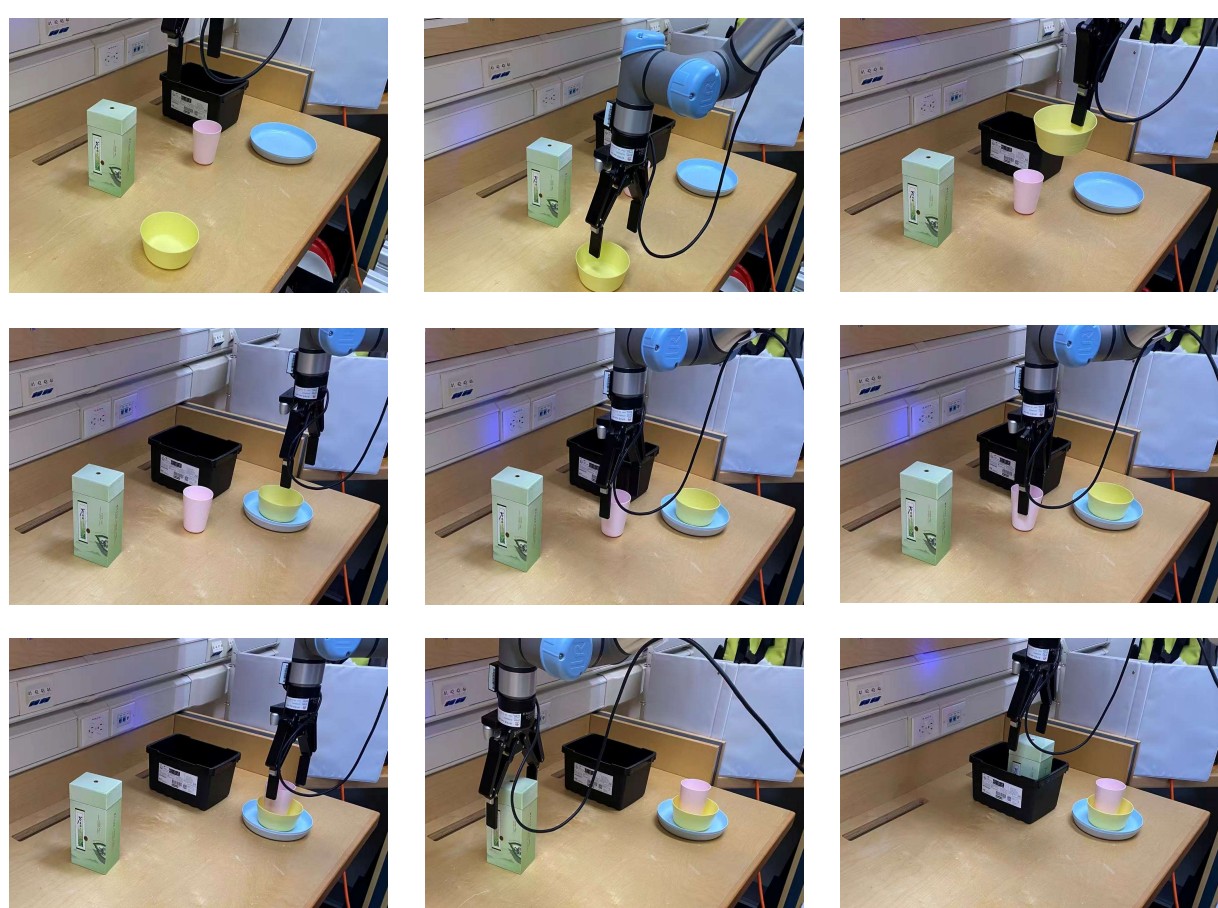

*Figure 6.* SCT-4B performing room organization task together with $\pi_0$

