# OpenReview forum: "Self-CriTeach: LLM Self-Teaching and Self-Critiquing for Improving Robotic Planning via Automated Domain Generation"
_ICML.cc/2026/Conference — ICML 2026 regular_

### Official Review · Reviewer_Df1u · 2026-03-07

**Soundness:** 3
**Presentation:** 3
**Significance:** 3
**Originality:** 3
**Overall Recommendation:** 4
**Confidence:** 2

**Summary:**

This paper proposes Self-CriTeach, a framework that generates Planning Domain Definition Language (PDDL) rules from demonstration data, and improves LLM planning capacity based on symbolic solutions of PPDL rules.

When generating data for self-teaching, the authors use Agilex Pika Data Collection System to collect human demonstration trajectories. To generate reliable PDDL rules, they introduced a closed-loop domain correction procedure that improves PDDL based on validation outcomes on simple tasks. The authors also employ a hill-climbing algorithm to remove redundant components. This procedure yields compact and feasible planning domains. With the planning domain, they generate problem-plan pairs by solving planning problems with a symbolic planner. The problem-plan pairs are transformed into chain-of-thought (CoT) traces with LLM.

LLMs are then trained on the automatically generated planning CoT data. The finetuning is a typical two-stage process: Supervised Fine-tuning (SFT) followed by Reinforcement Learning (RL). Self-CriTeach improves the success rate over 20% over the base LLM. Notably, their 4B model can outperform GPT-4o on Blocks World Hard.

**Compliance With Llm Reviewing Policy:**

Affirmed.

**Final Justification:**

My concerns are addressed.

**Key Questions For Authors:**

1. What if the collected human demonstration trajectories contain noise and suboptimal actions? Does that lead to bad PDDL results and degrade the performance of Self-CriTeach? Is Self-CriTeach sensitive to the quality of human demonstration?
2. The paper leaves the detail of closed-loop domain correction procedure to the appendix. Could you clarify some statistics of this procedure? For example, the success rate of generating valid PDDL domains, the number of loops required, and the stopping condition for the loop.
3. Could you discuss the limitation of PDDL, in particular the expressive capacity for more complicated robot tasks?

**Limitations:**

Discussed in the weakness part.

**Strengths And Weaknesses:**

**Strength**
1. The paper introduces an automatic process to generate CoT trace without expensive manual annotation.
2. The proposed Self-CriTeach method is empirical effective and show promising performance comparing to industry level model (GPT-4o).
3. The research question is important and the paper provides sound solution to it.
4. Self-CriTeach works in both simulation environment and real world robotic tasks.

**Weakness**
1. I suspect that the outcome of Self-CriTeach depends heavily on the initial demonstration trajectories and the expressive capacity of PDDL. It's not clear if Self-CriTeach still works well in longer horizon and more complicated tasks.
2. Self-CriTeach experiments only run on a single model size and architecture (Qwen3-4B). It would be interesting to see if Self-CriTeach works well for general model sizes. I understand that it's common nowadays for LLM reasoning/planning papers to only test their method on a single model size. But I still encourage the authors to try larger models.

---

> ### Author Rebuttal · Authors · 2026-03-30
>
> Thank you for your thoughtful review. We appreciate your recognition of our solution’s effectiveness, its strong performance, and its successful sim-to-real deployment. Below, we address your concerns.
>
> ---
> ## W1 & Q2
> **Stability of PDDL domain generation is ensured by a feedback-correction loop, enabling reliable downstream fine-tuning:** We acknowledge that reliable domain generation for complex, long-horizon tasks is more challenging. To address this, we integrate multiple feedback-based repair methods to improve robustness, including error trace-back prompting [1] and hill-climbing search [2].
>
> We evaluate domain generation success rates and average self-correction loops on Blocksworld tasks (stacking, unstacking, rearrangement, alignment), each over 10 trials. Domain generation is terminated after 5 self-correction loops and marked as failure. As shown in Table 4.1, consistently high success rates demonstrate strong practical stability under feedback-based correction.
>
> *Table 4.1: Domain generation success rates*
> | |Stacking|Unstacking|Rearrangement|Alignment|
> |-|-|-|-|-|
> |Success Rate|0.9|1.0|0.8|1.0|
> |Avg. Loops|1.7|1.3|2.4|1.5|
>
> ---
>  ## W1 & Q1
> **PDDL Domain generation remains relatively reliable under suboptimal and noisy demonstrations:** To evaluate robustness to noise, we mask a portion of the demonstration trajectory to simulate suboptimal demonstrations with one missing action (e.g., omitting either pick or stack in stacking tasks), while keeping all other experimental settings unchanged. Table 4.2 shows that incompleteness and noise increase domain correction effort and reduce overall success rate, but do not directly cause failure. In practice, accurate domains can still be ensured through parallel generation and selection.
>
> *Table 4.2: Domain generation success rates with incomplete demonstrations*
> | |Stacking|Unstacking|Rearrangement|Alignment|
> |-|-|-|-|-|
> |Success Rate|0.8|0.8|0.6|0.9|
> |Avg. Error-Correction Loops|2.8|2.6|3.6|2.5|
>
> **Self-CriTeach is resistant to minor PDDL domain inaccuracies:** Though we have shown the stability of PDDL domain generation in our system, we further examine the effect of domain inaccuracies by intentionally injecting errors into the domain model. Their impact depends on the type of error: minor inconsistencies are mostly tolerable, while critical errors can degrade performance by inducing flawed planning dynamics. We conduct two ablations on BW-Classic: (1) A minor inconsistency, using different names for the same predicate (graspable vs. clear) in 10% of the data, only slightly reduces success rate from 60% to 58%. (2) A critical error, removing the graspable precondition from the pick action and allowing objects to be picked while others remain on top, leads to a drop to 31%.
>
> ---
> ## W1 & Q3
> **Expressiveness Limitation of PDDL Domain for complex and long tasks:** PDDL struggles to model complex continuous physical dynamics, but excels at long-horizon planning. PDDL is well-suited for robotic tasks with deterministic transitions and well-defined geometry (e.g., rigid object manipulation [3]), but struggles to capture complex dynamics and fine-grained geometry (e.g., fluids or deformable objects). At the same time, PDDL's abstract representation is a strength for long-horizon planning, as it enables efficient reasoning over logical state transitions without modeling low-level details.
>
> **How PDDL work under challenging scenarios:** One promising future direction to address expressivity limitations is to combine PDDL-based planning with learned low-level controllers, such as vision-language-action (VLA) models. The PDDL-based planner handles long-term plan consistency and the VLA can handle rich perception and complex continuous dynamics. We have taken a preliminary step by integrating SCT-4B with a VLA model for real-world deployment, as described in Appendix A.8, suggesting a promising pathway for bridging symbolic planning and embodied control.
>
> ---
> ## W2
> **Self-CriTeach generalizes across models.** To verify that our framework is not tied to a specific model size, we apply Self-CriTeach to two additional 8B-scale LLMs, Llama-3.1-8B and Qwen3-8B. Both show consistent improvements. In particular, Llama-3.1-8B begins with a reasonable progress score but a low success rate, due to difficulty in tracking state transitions beyond a few planning steps. This limitation is largely mitigated after fine-tuning, resulting in a substantial gain in success rate.
>
> *Table 4.3: Model planning success rate and progress score (before vs. after SCT training)*
> |Model|Overall Success Rate|Overall Progress Score|
> |-|-|-|
> |SCT-Qwen3-8B|0.49|0.79|
> |Qwen3-8B|0.35|0.68|
> |SCT-Llama-8B|0.277|0.716|
> |Llama-3.1-8B|0.003|0.512|
>
>
> [1] Oswald et al., Large Language Models as Planning Domain Generators, ICAPS 2024
>
> [2] Silver et al., Predicate Invention for Bilevel Planning, AAAI 2023
>
> [3] Garrett et al., Integrated Task and Motion Planning, arxiv, 2020

---

> > ### Author Rebuttal · Reviewer_Df1u · 2026-04-01
> >
> > I appreciate the response from the author(s). I keep my positive score.

---

> > > ### Author Response · Authors · 2026-04-07
> > >
> > > Thank you very much for your thoughtful review of our paper. We truly appreciate your positive assessment of our work and your recognition of its novelty and technical soundness. We are also very grateful for the time and care you took in reading and acknowledging our rebuttal.

---

### Official Review · Reviewer_Ae84 · 2026-03-12

**Soundness:** 3
**Presentation:** 4
**Significance:** 4
**Originality:** 4
**Overall Recommendation:** 4
**Confidence:** 4

**Summary:**

The authors introduce Self-CriTeach, a framework where a model autonomously generates symbolic planning domains to serve as both a source of training data and an evaluation metric. The system uses these self-generated domains to automatically create accurate planning trajectories for supervised fine-tuning, and then applies the same domains as step-by-step reward signals for reinforcement learning. This unified training approach results in an enhanced model that demonstrates higher planning success rates, better generalization to unseen tasks, reduced computational costs, and improved robustness when deployed on real robots.

**Compliance With Llm Reviewing Policy:**

Affirmed.

**Final Justification:**

Major concerns are resolved by the author rebuttal, recommend accepting the paper.

**Key Questions For Authors:**

Q1: You mention in the limitations that the method might struggle with complex continuous constraints because it relies on symbolic logic. How difficult would it be to connect this purely symbolic reasoning approach with continuous action parameters, such as varying grip strength or continuous motion paths? The evidence or analysis showing that the error-correction steps—like the feedback prompting and hill-climbing search—are robust enough to recover from a poorly initialized domain will raise the confidence in the system's reliability.

Q2: You mention in the limitations that the method might struggle with complex continuous constraints because it relies on symbolic logic. How difficult would it be to connect this purely symbolic reasoning approach with continuous action parameters?

**Limitations:**

The authors have adequately discussed the limitations and potential negative societal impact of their work.

**Strengths And Weaknesses:**

The paper presents a solid and useful method for teaching language models to plan robot tasks without needing expensive human data. The core idea of using self-generated symbolic domains to create both training examples and reward signals is clever and well-tested. The authors run strong experiments, proving that their small 4-billion parameter model beats much larger models, including GPT-4o, on several planning tasks. They also provide clear ablation studies that prove every piece of their training pipeline is necessary to get the best results. Furthermore, testing the model on a real robot shows it can handle real-world perception noise better than traditional solvers.

However, the paper has a few limits that keep it from getting a perfect 6. The entire system relies heavily on the base model being smart enough to write an accurate starting domain. If the base model fails at this first step, the whole training process will likely break down. Additionally, the approach only works for tasks that can be perfectly translated into simple logic symbols, which means it will struggle with tasks that require complex, continuous physical control.

The paper presents a technically sound framework for improving robotic task planning. The experimental design is rigorous, evaluating the model on both seen and unseen tasks to test generalization capabilities, and comparing it against strong baselines of varying sizes. The ablation studies effectively isolate the contributions of the supervised fine-tuning (SFT), reinforcement learning (RL), and symbolic-to-Chain-of-Thought (CoT) components, proving that their combination yields the best results. Furthermore, testing the model on a real UR5e robot under perception noise validates its practical robustness compared to traditional symbolic solvers. However, the framework assumes the base LLM is already capable enough to generate an accurate initial PDDL domain, which is a potential point of failure for the entire pipeline. Additionally, the real-world experiments are limited to only 10 trials per task, which slightly restricts the statistical confidence of the real-world claims.

The paper is well-structured and the narrative is logically organized.

---

> ### Author Rebuttal · Authors · 2026-03-30
>
> Thank you for your thoughtful review. We appreciate your recognition of the novelty of our framework, the soundness of our experimental design, and the effectiveness of our ablation studies. We also thank you for acknowledging the practical robustness demonstrated in our real-robot experiments. Below, we address your concerns in detail.
>
> ---
> ## Q1 & Q2
> **Bridging symbolic planning with continuous parameters.** Bridging symbolic reasoning with continuous action parameters is a widely studied problem in robotics[1]. Existing work mainly follows two directions. The first uses foundation models to infer or refine pre-designed motion-level constraints from simulation or real-world feedback, where LLMs or VLMs generate code-like specifications such as grasp poses or waypoints for motion planners [2,3]. While training-free, these approaches remain limited for complex geometries such as deformable objects or fluids. The second direction uses learned policies that condition on symbolic actions to generate continuous robot control [4]. These methods better handle complex geometries, but typically require substantial training data. We have explored the compatibility of our method with both directions: Appendix A.7 demonstrates how we predesign motion-planner constraints can be integrated into a controller, and Appendix A.8 presents an integration with a Vision-Language-Action (VLA) model.
>
> ---
> ## Q1
>
> **PDDL domain generation is overall stable:** Self-correction plays a critical role in ensuring the correctness of PDDL domains. To validate domain generation stability, we evaluate success rates and the average number of self-correction loops on Blocksworld tasks, including stacking, unstacking, rearrangement, and alignment, with each setting tested over 10 trials. Domain generation is terminated and marked as failure if the LLM cannot correct the domain within 5 iterations.
>
> As shown in Table 3.1, domain generation achieves consistently high success rates. Most domains require only 1–2 self-correction loops, indicating that feedback prompting and hill-climbing together effectively ensure PDDL domain correctness. Although occasional failures are observed, they can be mitigated by parallel generation and selecting valid outputs.
>
> *Table 3.1: Domain generation success rates*
>
> | Metric                         | Stacking | Unstacking | Rearrangement | Alignment |
> |--------------------------------|----------|------------|---------------|-----------|
> | Success Rate               | 0.9      | 1.0        | 0.8           | 1.0       |
> | Avg. Error-Correction Loops| 1.7      | 1.3        | 2.4           | 1.5       |
>
> **PDDL domain generation remains reliable even under inaccurate initialization:** To evaluate robustness to inaccurate initialization, we intentionally mask part of the demonstration trajectory to cause one missing action in the initial domain (e.g., omitting either pick or stack in stacking tasks), while keeping all other settings unchanged. Table 3.2 shows that poorly initialized PDDL domains increase correction effort and lower success rate, but do not directly cause failure. In practice, accurate domains can still be ensured through parallel generation and selection.
>
> *Table 3.2: Domain generation success rates with poor initialization*
>
> |        | Stacking | Unstacking | Rearrangement | Alignment |
> |--------|----------|------------|---------------|-----------|
> | Success Rate | 0.8 | 0.8 | 0.6 | 0.9 |
> | Avg. Error-Correction Loops | 2.8 | 2.6 | 3.6 | 2.5 |
>
> ---
> ## Weakness
> While we acknowledge that requiring an LLM capable of generating a PDDL domain is a limitation, the results in Table 3.1 already demonstrate that such domains can be generated reliably, especially when recent feedback-based techniques are applied. Moreover, the rapid progress of modern models makes this prerequisite increasingly less restrictive in practice.
>
> We perform 10 additional real-robot experiments for each task under the same setting as in the original paper, resulting in a total of 20 trials per task. The updated results are shown in Table 3.3. Although we observe slight fluctuations compared with the 10-trial results, the overall trends remain consistent.
>
> *Table 3.3: Real-robot task success rates comparison between SCT-4B and PDDL Solver under different perception methods.*
> | Logical States Estimation | SCT-4B (Room) | SCT-4B (Lab) | PDDL Solver (Room) | PDDL Solver (Lab) |
> |-|-|-|-|-|
> | Qwen3-VL-4B | 0.75 | 0.60 | 0.45 | 0.20 |
> | Rule-based Classifier | 0.85 | 0.85 | 0.70 | 0.65 |
>
> [1] Garrett et al., *Integrated Task and Motion Planning*, arxiv, 2020
>
> [2] Liang et al., *VisualPredicator: Learning Abstract World Models with Neuro-Symbolic Predicates for Robot Planning*, ICLR, 2025
>
> [3] Han et al., *InterPreT: Interactive Predicate Learning from Language Feedback for Generalizable Task Planning*, RSS, 2024
>
> [4] Dalal et al., *Imitating Task and Motion Planning with Visuomotor Transformers*, CoRL 2023

---

> > ### Author Rebuttal · Reviewer_Ae84 · 2026-04-03
> >
> > Most of my concerns have been resolved. I will keep my rating positive.

---

> > > ### Author Response · Authors · 2026-04-07
> > >
> > > Thank you very much for your thoughtful review of our paper. We sincerely appreciate your positive evaluation of our work and your recognition of its novelty and technical soundness. We are also very grateful for the time and care you devoted to reading and acknowledging our rebuttal.

---

### Official Review · Reviewer_gXKc · 2026-03-14

**Soundness:** 3
**Presentation:** 2
**Significance:** 4
**Originality:** 4
**Overall Recommendation:** 5
**Confidence:** 3

**Summary:**

The paper proposes to leverage the planning domain generation ability of LLMs for finetuning. It argues that the symbolic domain model can

1. allow generation of problem-plan pairs that can be used for supervised fine-tuning,

2. the symbolic domain model can be leveraged to provide dense rewards for RL finetuning.

Empirical evaluation on blocks-world, hard and align variants of blocks world, and few unseen tasks suggest that the fine-tuning is helpful.

The proposed approach is termed as self-criteach, as it relies on LLM for generating the domain model, which is used for translation/curation of explanations to generate SFT training dataset [Teach part]. Further reward is also provided based on the LLM generated domain model [Critic part].

**Compliance With Llm Reviewing Policy:**

Affirmed.

**Final Justification:**

Authors address my concerns. Addition of LCCS baseline in the main paper would significantly improve the contribution as it provides a bridge to other reward-based finetuning. I would request authors to add that to the paper.

**Key Questions For Authors:**

Q1: Was the model only trained on Blockworld Classic, Hard, and Align? Not on any of the three unseen tasks? Could you describe how different/similar are the new domains to blocks world? I see the appendix has a list of objects that were added to this domain, but not sure how much variation that brings to domain dynamics or PDDL actions.

Q2: I am a little surprised to see Qwen3-8b performing better than GPT4o on classical Blockworld task. Any idea why is that so? I believe GPT4o is significantly larger model. How would the numbers look with the latest frontier models?

Q3: If the authors were to fine-tune with an “oracle” domain model (not generated by LLM), what difference would it make in the results. How much impact can noise or inaccurate domain model have on finetuning.

Q4: Since the main objective of the proposed approach is to improve plan-generation ability, other RL finetuning approaches used for the same would be a necessary baseline to understand the significance of this approach over others. See [1] for example of how LCCS can be used as dense reward for this purpose.



[1] Huang, S., Cohn, T., & Lipovetzky, N. (2025, September). Chasing progress, not perfection: Revisiting strategies for end-to-end llm plan generation. In Proceedings of the International Conference on Automated Planning and Scheduling (Vol. 35, No. 1, pp. 204-212).

**Limitations:**

Yes

**Strengths And Weaknesses:**

**Strengths**
- The problem setting is relevant, and important
- Proposed approach seems sound (albeit some details are missing)
- Approach is scalable and experiments on blocksworld and robotic domain suggest effectiveness


**Weaknesses**
- Does not compare other RL-finetuning approaches for plan-generation
- Paper lacks clarity on some details (see questions)

---

> ### Author Rebuttal · Authors · 2026-03-30
>
> Thank you for your thoughtful review. We appreciate your recognition of our solution’s effectiveness and the successful deployment. Below, we address your concern.
>
> ---
> ## Q1
> SCT-4B is trained only on Blocksworld Classic, Hard, and Align, and evaluated zero-shot on unseen scenarios. The unseen scenarios differ in three aspects:
> 1. Unseen Objects: Diverse unseen objects and furniture across environments.
> 2. Unseen Environment Layouts and Goals: Rather than a single-table setup, objects are placed across multiple pieces of furniture with novel spatial configurations and goal states.
> 3. Diverse Action Semantics: Instead of a fixed skill set, we allow varied but related action semantics across environments.
>
> ---
> ## Q2
> We answer this question with both theoretical explanation and detailed empirical failure analysis.
>
> **Qwen3-8B's training pipeline aligns with planning:** The Qwen3 series have a dedicated pretraining stage on math and code data [1], which benefits code-like planning structure such as PDDL [2]. Moreover, Qwen3-8B, trained with CoT distilled from stronger models, tends toward reasoning-oriented inference, enabling explicit step-wise planning. This is critical for checking sequential states, preconditions, and transitions in planning.
>
> **Empirical failure mode analysis.** We analyze failures on BW Classic, BW Hard, and BW Align and find the major failure mode to be precondition violations (over 97% of failures).
>
> *Table 2.1: Success rate of models across plan length.*
> |ground-truth plan length|Qwen3-8B|GPT-4o|
> |-|-|-|
> |Short (≤8 steps)|63.2%|61.7%|
> |Medium (8–12 steps)|35.2%|12.6%|
> |Long (>12 steps)|11.2%|7.4%|
>
> **Model performances diverge in medium-length plans:** Table 2.1 shows both models achieve similar success rates on short tasks, with a clear gap emerging in medium tasks. This supports our hypothesis that Qwen3-8B's step-by-step state tracking reduces constraint violations. In contrast, GPT-4o’s direct-answer preference leads to lower success due to poor global consistency. For long tasks, Qwen3-8B also drops to 11.2%, likely reflecting limitations due to model capacity.
>
> **Frontier model performance:** As a reference, Qwen3.5-27B achieves 99% success on short and medium tasks and 88% on long tasks across three Blocksworld variants, with an average token cost of 14,826. SCT-4B achieves competitive results despite being ~1/8 the size and ~1/2 token cost, showing our pipeline as a strong alternative to scaling, with advantages in lightweight design and low token cost that are important for robotics.
>
> ---
> ## Q3
> **Performances are similar with oracle domains:** Recent studies show that automatically generated PDDL domains can approach human-designed quality [3–4]. Thus, models fine-tuned on oracle domains(SCT-4B-Oracle) exhibit similar performance to SCT-4B, as is shown in Table 2.2.
>
> *Table 2.2: Performance of model fine-tuned with oracle PDDL domain.*
> |Training Domain|Seen SR/Progress|Unseen SR/Progress|Overall SR/Progress|
> |-|-|-|-|
> | SCT-4B |60%/85%|38%/73%|46%/76%|
> | SCT-4B-Oracle|57%/82%|39%/73%|45%/78%|
>
> **The influence of inaccuracies & noise depends on their criticality:** Though noisy and inaccurate domains become rare with recent advances [3–4], their impact depends on the nature of the error. Minor errors are tolerable, while key errors degrade performance by inducing flawed planning dynamics. We conduct two ablations on BW-Classic with intentionally perturbed domain models. (1) A minor inconsistency, using different names for the same predicate (graspable vs. clear) in 10% of the data, has only a small impact, with 58% success compared to 60% for a consistent domain. (2) A critical error, removing the graspable precondition from the pick action and allowing objects to be picked while others remain on top, causes a large drop in success rate to 31%.
>
> ---
> ## Q4
> We fine-tuned Qwen3-4B using the LCCS reward defined in the referenced paper. The results are reported in Table 2.3:
>
> *Table 2.3: Performance of LCCS Baseline.*
> |BW-Classic SR|BW-Hard SR|BW-Align SR|Prepare Experiment SR|Reorganize Room SR|Machine Assembly SR|Overall SR|Progress Score|
> |-|-|-|-|-|-|-|-|
> |49%|36%|51%|25%|17%|30%|31%|71%|
>
> LCCS outperforms DPO in both success rate(SR) and progress score, and matches CPO in success rate (31%) while achieving higher progress (71% vs. 69%), indicating that optimizing for longer-horizon progress improves partial plan generation. However, it remains below the full SCT pipeline, which leverages the planning domain as both a reward signal and a supervision source.
>
> [1] Yang et al., Qwen3 Technical Report, arxiv,2025
>
> [2] Pallagani et al., Understanding the Capabilities of Large Language Models for Automated Planning, arxiv,2023
>
> [3] Liang et al., VisualPredicator: Learning Abstract World Models with Neuro-Symbolic Predicates for Robot Planning, ICLR,2025
>
> [4] Han et al., InterPreT: Interactive Predicate Learning from Language Feedback for Generalizable Task Planning, RSS,2024

---

> > ### Author Rebuttal · Reviewer_gXKc · 2026-04-01
> >
> > Thanks you for addressing my concerns. I will update my score accordingly.

---

> > > ### Author Response · Authors · 2026-04-07
> > >
> > > Thank you very much for your thoughtful review of our paper and for your consideration during the rebuttal process. We sincerely appreciate your encouraging assessment of our work and your recognition of its novelty and technical soundness. We are also very grateful for the time and care you devoted to reading our rebuttal.

---

### Official Review · Reviewer_r33s · 2026-03-15

**Soundness:** 3
**Presentation:** 3
**Significance:** 3
**Originality:** 3
**Overall Recommendation:** 4
**Confidence:** 3

**Summary:**

This paper proposes Self-CriTeach, a method to let LLM teach its own CoT planning trajectory and rewards via self-generated PDDL program. The idea is intuitive and the results demonstrated to be effective. Although it depends on strong enough base LLM to generate good enough PDDL, but that is not a true limitation these days.

**Compliance With Llm Reviewing Policy:**

Affirmed.

**Final Justification:**

The added experiments addresses my concerns to compare with vlm+pddl and shows it is better. However, the sample number is small and the difference is not that huge. The limitation for continuous state is explicitly acknowledged.

**Key Questions For Authors:**

none

**Limitations:**

1. PDDL program quality limited by base model.
2. Continuous state suffers from symbolic planning.

**Strengths And Weaknesses:**

Strength
1. Intuitive idea - the ability to generate a program is not same as the ability to internalize the run-time dynamics of the program. It is a source of self-improving.
2. The experiments covers a wide range of situations - stronger model, distill, RL, SFT, domain generalization.

Weakness
1. the major experiment that I want to see is to compare with test-time PDDL augmentation.  I know in the real world experiments it shows classical PDDL can't handle noisy signal well, but a better comparison is to combine VLM/VLA with PDDL so that the noise is handled by the large model and it still benefits from PDDL.

---

> ### Author Rebuttal · Authors · 2026-03-30
>
> Thank you for your thoughtful review and for your patience. We appreciate your recognition of the novelty of our framework and the effectiveness of the training outcomes. Below, we address each concern in detail.
>
> ## Weakness 1
>
> We agree that test-time PDDL augmentation with foundation models (e.g., VLMs) is an important technique. Our approach instead learns an end-to-end system that internalizes planning behavior and achieves implicit resistance to noise, reducing engineering effort for such multi-system integration.
>
> **VLM-based PDDL state estimation refinement also benefits our approach**: Moreover, we view these test-time PDDL augmentation approaches as complementary to ours, as logical state-refinement techniques could further benefit our method by providing more accurate input logical state estimates. To this end, we include an additional experiment that integrates a VLM-based online PDDL state refinement module with both a classical PDDL solver (denoted as VLM Aug. + PDDL) and our SCT-4B model (denoted as VLM Aug. + SCT-4B). The VLM used is Qwen3-VL-4B, and 10 trials were conducted for each setting to compute the success rate. During VLM augmentation, we imitate the real-robot experimental pipeline in [1], where the VLM is provided with the current observation, failure signals, error trace, and task-specific PDDL domain semantics to perform refinement. For each trial, a maximum of two refinement iterations is allowed.
>
> *Table 1.1: Reorganize Room — real-robot experiment results under VLM-based PDDL state augmentation.*
> | Logical States Estimation | SCT-4B | VLM Aug. + SCT-4B | PDDL Solver | VLM Aug. + PDDL |
> |--------------------------|--------|--------------------|-------------|------------------|
> | Qwen3-VL-4B              | 0.70   | 0.90               | 0.40        | 0.70             |
> | Rule-based Classifier    | 0.80   | 0.80               | 0.70        | 0.80             |
>
>
> *Table 1.2: Prepare Experiment — real-robot experiment results under VLM-based PDDL state augmentation.*
> | Logical States Estimation | SCT-4B | VLM Aug. + SCT-4B | PDDL Solver | VLM Aug. + PDDL |
> |--------------------------|--------|--------------------|-------------|------------------|
> | Qwen3-VL-4B              | 0.60   | 0.70               | 0.20        | 0.20             |
> | Rule-based Classifier    | 0.90   | 0.90               | 0.70        | 0.80             |
>
>
> **The improvement from VLM-based PDDL augmentation varies across tasks**: Tables 1.1 and 1.2 show that VLM-based PDDL refinement benefits both SCT-4B and the PDDL solver, although the impact varies across tasks. In Table 1.1 (*Reorganize Room*), where objects are visually distinct in color and have relatively simple geometry, VLM augmentation provides substantial gains, particularly for the PDDL solver. In contrast, in Table 1.2 (*Prepare Experiment*), where objects exhibit complex geometry and partial transparency, VLM-based state estimation struggles with spatial relationships, resulting in limited performance improvement. We further observe from Table 1.1 (Reorganize Room) that, when logical states are initialized by rule-based classifiers, VLM Aug. + SCT-4B achieves a similar success rate to standalone SCT-4B. Analysis suggests that the failures mainly come from corner cases in the rule-based classifier, which produces incorrect object relations. This is a known limitation in classical task and motion planning and can mislead the subsequent VLM-based correction loop. The same issue also affects VLM Aug. + PDDL, causing both methods to converge to a similar success rate of 0.8.
>
> ---
>
> ## Limitations
>
> We appreciate your point that, with recent advances in foundation models, dependence on a sufficiently capable base model is becoming less restrictive in practice. Recent work on feedback-based PDDL domain refinement also suggests that the reliability of domain generation can be further improved [2–3].
>
> We agree that interfacing symbolic reasoning with continuous parameters remains an open challenge and important future direction. As an initial step, Appendix A.8 includes an experiment validating the compatibility of our method with a Vision-Language-Action (VLA) model. We plan to investigate this direction further in future work.
>
>
>
> [1] Zhu et al., PSALM-V: Automating Symbolic Planning in Interactive Visual Environments with Large Language Models, arxiv, 2025
>
> [2] Liang et al., VisualPredicator: Learning Abstract World Models with Neuro-Symbolic Predicates for Robot Planning, ICLR, 2025
>
> [3] Han et al., InterPreT: Interactive Predicate Learning from Language Feedback for Generalizable Task Planning, RSS, 2024

---

### Decision · Program_Chairs · 2026-04-30

**Decision:**

Accept (regular)

**Comment:**

The paper presents an approach to generating training data for planning-problems, and for a structured reward mechanism using generated CoT trajectories.

The reviews are unanimous in agreeing that the proposed approach is technically sound, and promising. The results are comprehensive, and the ablations show the importance of the design decisions.

There were some interesting questions raised in the discussion, though - including the dependence on the capabilities of the base model, and why/how a 4B model is able to outperform a frontier model. I think the authors can address them in a discussion section in the revised paper.

Overall this paper looks like it would make a solid contribution to the field.